# Iterated $Q$-Network: Beyond One-Step Bellman Updates in Deep Reinforcement Learning

**Théo Vincent**                                                                     *theo.vincent@dfki.de*
*DFKI, SAIROL Team & TU Darmstadt*

**Daniel Palenicek**                                                    *daniel.palenicek@tu-darmstadt.de*
*TU Darmstadt & Hessian.ai*

**Boris Belousov**                                                       *boris.belousov@robot-learning.de*
*DFKI, SAIROL Team*

**Jan Peters**                                                              *peters@ias.tu-darmstadt.de*
*DFKI, SAIROL Team & TU Darmstadt & Hessian.ai*

**Carlo D'Eramo**                                                        *carlo.deramo@uni-wuerzburg.de*
*University of Würzburg & TU Darmstadt & Hessian.ai*

**Reviewed on OpenReview:** *https://openreview.net/forum?id=Lt2H8Bd8jF*

## Abstract

The vast majority of Reinforcement Learning methods is largely impacted by the computation effort and data requirements needed to obtain effective estimates of action-value functions, which in turn determine the quality of the overall performance and the sample-efficiency of the learning procedure. Typically, action-value functions are estimated through an iterative scheme that alternates the application of an empirical approximation of the Bellman operator and a subsequent projection step onto a considered function space. It has been observed that this scheme can be potentially generalized to carry out multiple iterations of the Bellman operator at once, benefiting the underlying learning algorithm. However, until now, it has been challenging to effectively implement this idea, especially in high-dimensional problems. In this paper, we introduce *iterated Q-Network* (i-QN), a novel principled approach that enables multiple consecutive Bellman updates by learning a tailored sequence of action-value functions where each serves as the target for the next. We show that i-QN is theoretically grounded and that it can be seamlessly used in value-based and actor-critic methods. We empirically demonstrate the advantages of i-QN in Atari 2600 games and MuJoCo continuous control problems. Our code is publicly available at *https://github.com/theovincent/i-DQN* and the trained models are uploaded at *https://huggingface.co/TheoVincent/Atari_i-QN*.

## 1 Introduction

Deep Reinforcement Learning (RL) algorithms have achieved remarkable success in various fields, from nuclear physics (Degrave et al., 2022) to construction assembly tasks (Funk et al., 2022). These algorithms aim at obtaining a good approximation of an action-value function through *consecutive* applications of the Bellman operator $\Gamma$ to guide the learning procedure in the space of $Q$-functions, i.e., $Q_0 \rightarrow \Gamma Q_0 = Q_1 \rightarrow \Gamma Q_1 = Q_2 \rightarrow \cdots$ (Bertsekas, 2019). This process is known as *value iteration*. *Approximate value iteration* (AVI) (Farahmand, 2011) extends the value iteration scheme to function approximation by adding a projection step, $Q_0 \rightarrow \Gamma Q_0 \approx Q_1 \rightarrow \Gamma Q_1 \approx Q_2 \rightarrow \cdots$. Thus, the $k^{\text{th}}$ Bellman update $\Gamma Q_{k-1}$ gets projected back onto the chosen $Q$-function space via a new function $Q_k$ approximating $\Gamma Q_{k-1}$. This projection step

also appears in *approximate policy evaluation* (APE), where an empirical version of the Bellman operator for a behavioral policy is repeatedly applied to obtain its value function (Sutton & Barto, 1998).

In this paper, we discuss and tackle two efficiency issues resulting from the AVI learning scheme. **(i)** Projection steps are made sequentially in the sense that the following Bellman update is only considered by the learning process once the current projection step is frozen. For example, $\Gamma Q_1$ starts to be learned only once $Q_1$ is frozen. This harms the efficiency of the training. **(ii)** Samples are only used to learn a *one-step* application of the Bellman operator at each gradient step, reducing sample-efficiency. We propose a novel approach to overcome these limitations by *learning consecutive Bellman updates simultaneously*. We leverage neural network function approximation to learn consecutive Bellman updates in a telescopic manner, forming a chain where each

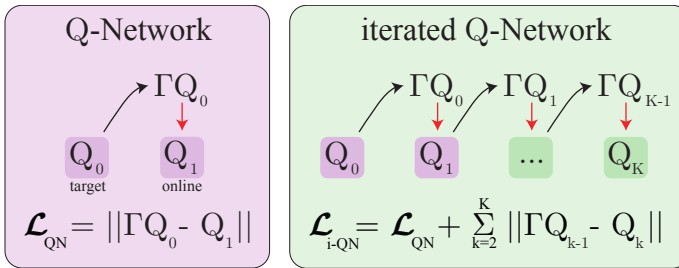

Figure 1: *Iterated Q-Network* (ours) uses the online network of regular *Q-Network* approaches to build a target for a second online network, and so on, through the application of the Bellman operator $\Gamma$. The resulting loss $\mathcal{L}_{\text{i-QN}}$ comprises $K$ temporal difference errors instead of just one as in $\mathcal{L}_{\text{QN}}$.

neural network learns the application of the Bellman operator over the previous one, as shown in Figure 1. This leads to a hierarchical ordering between the $Q$-estimates, where each one is the projection of the Bellman update corresponding to the previous one, hence the name *iterated Q-Network* (i-QN). Importantly, i-QN distributes the samples across all considered projection steps, thereby increasing the number of samples that each $Q$-function is learned from. Our approach can be seamlessly used in place of the regular one-step Bellman update by any value-based or actor-critic method, e.g., Deep $Q$-Network (DQN) (Mnih et al., 2015), Soft Actor-Critic (SAC) (Haarnoja et al., 2018). In the following, we motivate our approach theoretically and provide an algorithmic implementation that we validate empirically on Atari 2600 (Bellemare et al., 2013) and MuJoCo control problems (Todorov et al., 2012).

**Contributions.** (1) We introduce *iterated Q-Network* (i-QN), a novel approach that enables learning multiple Bellman updates at once. (2) We provide intuitive and theoretical justifications for the advantages and soundness of i-QN. (3) We show that i-QN can be seamlessly combined with value-based and actor-critic methods to enhance their performance, conducting experiments on Atari games and MuJoCo tasks.

## 2 Preliminaries

We consider discounted Markov decision processes (MDPs) defined as $\mathcal{M} = \langle \mathcal{S}, \mathcal{A}, \mathcal{P}, \mathcal{R}, \gamma \rangle$, where $\mathcal{S}$ and $\mathcal{A}$ are measurable state and action spaces, $\mathcal{P} : \mathcal{S} \times \mathcal{A} \to \Delta(\mathcal{S})$[1] is a transition kernel, $\mathcal{R} : \mathcal{S} \times \mathcal{A} \to \Delta(\mathbb{R})$ is a reward function, and $\gamma \in [0, 1)$ is a discount factor (Puterman, 1990). A policy is a function $\pi : \mathcal{S} \to \Delta(\mathcal{A})$, inducing an action-value function $Q^\pi(s, a) \triangleq \mathbb{E}_\pi \left[ \sum_{t=0}^\infty \gamma^t \mathcal{R}(s_t, a_t) | s_0 = s, a_0 = a \right]$ that gives the expected discounted cumulative return executing action $a$ in state $s$, following policy $\pi$ thereafter. The objective is to find an optimal policy $\pi^* = \arg\max_\pi V^\pi(\,\cdot\,)$, where $V^\pi(\,\cdot\,) = \mathbb{E}_{a \sim \pi(\,\cdot\,)}[Q^\pi(\,\cdot\,, a)]$. Approximate value iteration (AVI) and approximate policy iteration (API) are two common paradigms to tackle this problem (Sutton & Barto, 1998). While AVI aims to find the optimal action-value function $Q^*(\cdot, \cdot) \triangleq \max_\pi Q^\pi(\cdot, \cdot)$, API alternates between approximate policy evaluation (APE) to estimate the action-value function of the current policy and policy improvement that improves the current policy from the action-value function obtained from APE.

Both paradigms aim to find the fixed point of a Bellman equation by repeatedly applying a Bellman operator starting from a random $Q$-function. AVI uses the *optimal Bellman operator* $\Gamma^*$, whose fixed point is $Q^*$, while APE relies on the *Bellman operator* $\Gamma^\pi$ associated with a policy $\pi$, whose fixed point is $Q^\pi$ (Bertsekas,

---

[1]$\Delta(\mathcal{X})$ denotes the set of probability measures over a set $\mathcal{X}$.

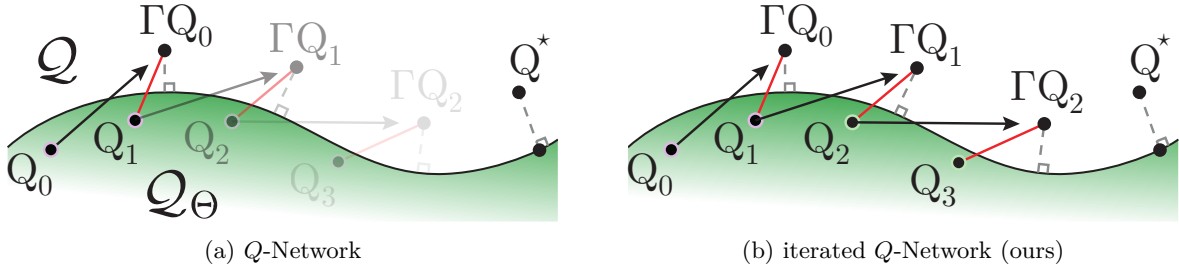

(a) $Q$-Network

(b) iterated $Q$-Network (ours)

Figure 2: Graphical representation of the regular $Q$-Network approach (left) compared to our proposed iterated $Q$-Network approach (right) in the space of $Q$-functions $\mathcal{Q}$. The regular $Q$-Network approach proceeds sequentially, i.e., $Q_2$ is learned only when the learning process of $Q_1$ is finished. With iterated $Q$-Network, all parameters are learned simultaneously. The projection of $Q^\star$ and projections of the Bellman update are depicted with a dashed line[2]. The losses are shown in red.

2015). For any state $s \in \mathcal{S}$ and action $a \in \mathcal{A}$, $\Gamma^*$ and $\Gamma^\pi$ are defined as

$$(\Gamma^* Q)(s, a) \triangleq \mathbb{E}_{r \sim \mathcal{R}(s,a), s' \sim \mathcal{P}(s,a)}[r + \gamma \max_{a' \in \mathcal{A}} Q(s', a')], \tag{1}$$

$$(\Gamma^\pi Q)(s, a) \triangleq \mathbb{E}_{r \sim \mathcal{R}(s,a), s' \sim \mathcal{P}(s,a), a' \sim \pi(s')}[r + \gamma Q(s', a')]. \tag{2}$$

It is well-known that these operators are contraction mappings in $L_\infty$-norm, such that their iterative application leads to their fixed point in the limit (Bertsekas, 2015). However, in model-free RL, only sample estimates of those operators are used since the expectations cannot be computed in closed form. This approximation, coupled with the use of function approximation to cope with large state-action spaces, forces learning the values of a Bellman update before being able to compute the next Bellman update. This procedure, informally known as *projection step*, results in a sequence of projected functions $(Q_i)$ that does not correspond to the one obtained from the consecutive applications of the true Bellman operator, as shown in Figure 2a, where $\Gamma$ equals $\Gamma^*$ for AVI and $\Gamma^\pi$ for APE, and $Q^\star$ represents $Q^*$ for AVI and $Q^\pi$ for APE. We denote $\mathcal{Q}_\Theta$ as the space of function approximators, where $\Theta$ is the space of parameters. To learn a Bellman update $\Gamma Q_{\bar{\theta}_0}$ given a fixed parameter vector $\bar{\theta}_0 \in \Theta$, for a sample $s, a, r, s'$, Temporal Difference (TD) learning (Hasselt, 2010; Haarnoja et al., 2018) aims to minimize

$$\mathcal{L}_{\text{QN}}(\theta_1 | \bar{\theta}_0, s, a, r, s') = \left( \hat{\Gamma} Q_{\bar{\theta}_0}(s, a) - Q_{\theta_1}(s, a) \right)^2, \tag{3}$$

over parameters $\theta_1 \in \Theta$, where $\hat{\Gamma}$ is an empirical estimate of $\Gamma$ and QN stands for $Q$-Network (Dayan & Watkins, 1992; Mnih et al., 2015). As an example, in a $Q$-learning setting, $\hat{\Gamma} Q(s, a) = r + \gamma \max_{a'} Q(s', a')$, for $Q \in \mathcal{Q}$ and a sample $(s, a, r, s')$. In current approaches, a single Bellman update is learned at a time. The training starts by initializing the target parameters $\bar{\theta}_0$ and the online parameters $\theta_1$. The distance between $Q_{\theta_1}$, representing the second $Q$-function $Q_1$, and $\Gamma Q_{\bar{\theta}_0}$, representing the first Bellman update $\Gamma Q_0$, is minimized via the loss in Equation 3, as shown in Figure 2a. After a predefined number of gradient steps, the procedure repeats, as in Figure 2a, where the second and third projection steps are blurred to stress the fact that they are learned sequentially. However, this sequential execution is computationally inefficient because it requires many *non-parallelizable* gradient steps to train a single projection. Moreover, this procedure is sample-inefficient since, at each gradient step, samples are used to learn only one Bellman update. In this work, we present a method that learns multiple Bellman updates at each gradient from a *single* sample batch. Importantly, our method scales to deep RL as shown in Section 6.

---

[2]The existence of a projection on $\mathcal{Q}_\Theta$ depends only on the choice of the function approximator. Note that even if the projection does not exist or if the function approximators are powerful enough to cover the entire space of $Q$-functions, i.e., $\mathcal{Q}_\Theta = \mathcal{Q}$, the presented abstraction is still valid. Indeed, in any case, applying the Bellman operator moves away the target from the $Q$-function. Therefore, learning Bellman updates is still required.

# 3 Related work

Several methods have been proposed on top of $Q$-learning to improve various aspects. A large number of those algorithms focus on variants of the empirical Bellman operator (Van Hasselt et al., 2016; Fellows et al., 2021; Sutton, 1988). For instance, double DQN (Van Hasselt et al., 2016) uses an empirical Bellman operator designed to avoid overestimating the return. As shown in Figure 3, this results in a different location of the Bellman update $\tilde{\Gamma}Q$ compared to the classical Bellman update $\widehat{\Gamma}Q$. Other approaches consider changing the space of representable $Q$-functions $\mathcal{Q}_\Theta$ (Wang et al., 2016; Osband et al., 2016; Fatemi & Tavakoli, 2022; Ota et al., 2021), attempting to improve the projection of $Q^\star$ on $\mathcal{Q}_\Theta$ compared to the one for the chosen

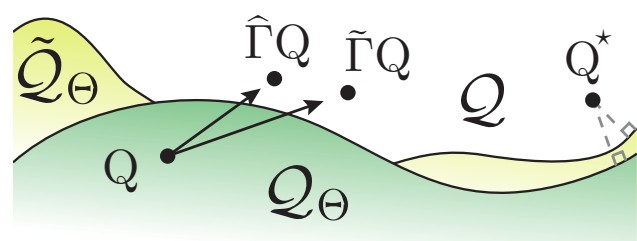

Figure 3: Other empirical Bellman operators can be represented using another notation $\tilde{\Gamma}$ than the classical empirical Bellman operator $\widehat{\Gamma}$. Changing the class of function approximators $\mathcal{Q}_\Theta$ results in a new space $\tilde{\mathcal{Q}}_\Theta$.

baseline's neural network architecture. It is important to note that adding a single neuron to one architecture layer can significantly change $\mathcal{Q}_\Theta$. Wang et al. (2016) show that performance can be increased by including inductive bias into the neural network architecture. This idea can be understood as a modification of $\mathcal{Q}_\Theta$, as shown in Figure 3 where the new space of representable $Q$-function $\tilde{\mathcal{Q}}_\Theta$ is colored in yellow. Furthermore, algorithms such as Rainbow (Hessel et al., 2018) leverage both ideas.

To the best of our knowledge, only a few works consider learning multiple Bellman updates concurrently. Using a HyperNetwork (Ha et al., 2016), Vincent et al. (2024) propose to approximate the $Q$-function parameters resulting from consecutive Bellman updates. While the idea of learning a HyperNetwork seems promising, one limitation is that it is challenging to scale to action-value functions with millions of parameters (Mnih et al., 2015). Similarly, Schmitt et al. (2022) provide a theoretical study about the learning of multiple Bellman updates concurrently. They consider an off-policy learning setting for linear function approximation and focus on low-dimensional problems. Crucially, their study shows that, in the limit, the concurrent learning of a sequence of consecutive $Q$-functions converges to the same sequence of $Q$-functions when learning is carried out sequentially. This finding leads to the following statement: *we do not need to wait until one Q-function has converged before learning the following one*. In this work, we pursue this idea by proposing a novel approach that scales with the number of parameters of the $Q$-function.

# 4 Learning multiple Bellman updates

The motivation behind learning multiple Bellman updates arises from a well-known result in AVI, which establishes an upper bound on the *performance loss* $\|Q^* - Q^{\pi_N}\|$, where $\pi_N$ is the greedy policy of $Q_N$, i.e., the last $Q$-function learned during training (Theorem 3.4 from Farahmand (2011), stated in Appendix A for completeness). The only term of this upper bound that is controlled by the optimization procedure is the *sum of approximation errors*[3] $\sum_{k=1}^{N} \|\Gamma^* Q_{k-1} - Q_k\|_{2,\nu}^2$, where $\nu$ is the distribution of the state-action pairs in the replay buffer. Therefore, minimizing this sum of terms is crucial to obtain low performance losses. Importantly, for $\theta, \bar{\theta} \in \Theta$, minimizing $\theta \mapsto \sum_{(s,a,r,s') \in \mathcal{D}} \mathcal{L}_{\mathrm{QN}}(\theta|\bar{\theta}, s, a, r, s')$ is equivalent to minimizing $\theta \mapsto \|\Gamma^* Q_{\bar{\theta}} - Q_\theta\|_{2,\nu}^2$ under the condition that the dataset of samples $\mathcal{D}$ is rich enough to represent the true Bellman operator as proven in Proposition A.1 of Appendix A. This means that by learning one Bellman update at a time, classical $Q$-Network approaches minimize only one approximation error at a time. Therefore, by optimizing for the immediate approximation error, classical $Q$-Network approaches can be seen as greedy algorithms aiming at minimizing the sum of approximation errors. Unfortunately, this does not guarantee a low sum of approximation errors at the end of the training as greedy algorithms can get trapped in local minima or even diverge with the number of iterations. This is why, in this work, we propose a novel approach that considers several consecutive Bellman updates simultaneously to aim at directly minimizing the sum of

---

[3]We ignore the factors $\alpha_k$ weighting the sum of approximation errors as they do not play a significant role in the analysis.

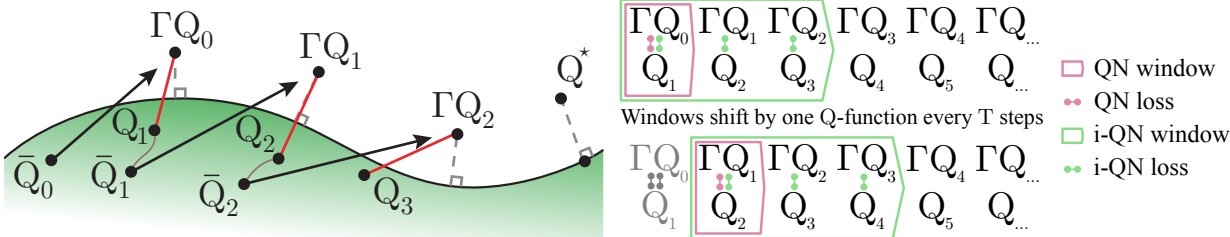

Figure 4: **Left:** Graphical representation of i-QN where each online networks $Q_k$ learns from its respective target network $\bar{Q}_{k-1}$. Every $G$ steps, each target network $\bar{Q}_k$ ($k > 0$) is updated to its respective $Q_k$. **Right:** i-QN considers a window of $K$ Bellman updates as opposed to QN methods that consider only 1 Bellman update. Every $T$ steps, the windows are shifted forward to consider the following Bellman updates.

approximation errors, as illustrated in Figure 2b. We complement this abstract visualization with Figure 10 in Appendix C.1, where we verify the described behavior in practice on a simple two-dimensional problem by applying our approach to Fitted-$Q$ iteration (Ernst et al., 2005).

To learn $K$ consecutive Bellman updates, we consider an ensemble of $K$ online parameters $(\theta_k)_{k=1}^K$. Each $Q$-function $Q_{\theta_k}$ is responsible for learning one Bellman update. Each Bellman update is learned from a target network so that the minimization of the TD-error leads to the minimization of the corresponding approximation error as previously discussed. This means that we also consider $K$ target parameters $(\bar{\theta}_k)_{k=0}^{K-1}$, where each online parameter $\theta_k$ is learned to fit its corresponding Bellman update computed from the target parameters $\bar{\theta}_{k-1}$. The target parameters are updated in two different ways to ensure that the considered networks are learning consecutive Bellman updates. First, to enforce the structure of a chain, we update each target network $\bar{\theta}_k$ ($k > 0$) to its corresponding online network $\theta_k$ every $D$ steps as Figure 4 (left) illustrates, in which we note $Q_k = Q_{\theta_k}$ and $\bar{Q}_k = Q_{\bar{\theta}_k}$ for clarity. Then, learning all the Bellman updates is not feasible in practice as it would require storing many $Q$-functions parameters. Instead, we learn a window composed of $K$ consecutive Bellman updates that is shifted every $T$ steps as shown in Figure 4 (right). The window is shifted by updating each target parameter $\bar{\theta}_k$ to the following online parameter $\theta_{k+1}$ (see Figure 4 (left)). Classical QN approaches also fit in this representation as they consider a window of size 1.

Unfortunately, while i-QN minimizes each approximation error by considering consecutive Bellman updates, the sum of approximation errors is not necessarily minimized. This is due to the semi-gradient update rule which creates gaps between each *frozen* target network $Q_{\bar{\theta}_k}$ and the online network $Q_{\theta_k}$ they represent as shown in Figure 4 (left). This means that even when each approximation error $\|\Gamma Q_{\bar{\theta}_{k-1}} - Q_{\theta_k}\|_{2,\nu}^2$ decreases by minimizing the corresponding QN loss between each $\theta_k$ and $\bar{\theta}_{k-1}$, i.e. $\sum_{k=1}^K \|\Gamma^* Q_{\bar{\theta}_{k-1}} - Q_{\theta_k}\|_{2,\nu}^2$ decreases, the sum of approximation errors $\|\Gamma^* Q_{\bar{\theta}_0} - Q_{\theta_1}\|_{2,\nu}^2 + \sum_{k=2}^K \|\Gamma^* Q_{\theta_{k-1}} - Q_{\theta_k}\|_{2,\nu}^2$ (without the target parameters) does not necessarily decrease.

Nevertheless, in the following, we derive a sufficient condition under which the considered sum of approximation errors is guaranteed to decrease, thus proving i-QN's soundness. For that, we focus on the timesteps where the gap between the target networks and the online networks they represent is null. This happens every $D$ timesteps as each target network $Q_{\bar{\theta}_k}$ is updated to its respective online network $Q_{\theta_k}$. Therefore, we note $(\theta_k^t)_{k=0}^K$ the value of the parameters the $t^{\text{th}}$ time the target parameters $(\bar{\theta}_k)_{k=1}^{K-1}$ are updated to their respective online parameters $(\theta_k)_{k=1}^{K-1}$. We stress that on those timesteps, the target parameters do not need to be distinguished from the online parameters anymore as they are equal. Proposition 4.1 states that between two timesteps $t$ and $t + 1$, the sum of approximation errors is guaranteed to decrease (Equation 5) if the decrease of each approximation error after optimization compared to their previous version before optimization is greater than a certain quantity: the displacement of the targets, i.e., $\|\Gamma^* Q_{\theta_{k-1}^{t+1}} - \Gamma^* Q_{\theta_{k-1}^t}\|_{2,\nu}$ (Equation 4). The proof, which relies mainly on the triangular inequality, is available in Appendix A.

---

**Algorithm 1** Iterated Deep Q-Network (i-DQN). Modifications to DQN are marked in purple.

---

1: Initialize the first target network $\bar{\theta}_0$ and the $K$ online parameters $(\theta_k)_{k=1}^K$, and an empty replay buffer $\mathcal{D}$. For $k = 1, .., K-1$, set $\bar{\theta}_k \leftarrow \theta_k$ the rest of the target parameters.
2: **repeat**
3:     Sample $k^b \sim U\{1, .., K\}$.
4:     Take action $a_t \sim \epsilon\text{-greedy}(Q_{\theta_{k^b}}(s_t, \cdot))$; Observe reward $r_t$, next state $s_{t+1}$.
5:     Update $\mathcal{D} \leftarrow \mathcal{D} \bigcup \{(s_t, a_t, r_t, s_{t+1})\}$.
6:     **every $G$ steps**
7:         Sample a mini-batch $\mathcal{B} = \{(s, a, r, s')\}$ from $\mathcal{D}$.
8:         **for** $k = 1, .., K$ **do** [*in parallel*]
9:             Compute the loss w.r.t. $\theta_k$, $\mathcal{L}_{\text{DQN}} = \sum_{(s,a,r,s') \in \mathcal{B}} \left( r + \gamma \max_{a'} Q_{\bar{\theta}_{k-1}}(s', a') - Q_{\theta_k}(s, a) \right)^2$.
10:             Update $\theta_k$ from $\nabla_{\theta_k} \mathcal{L}_{\text{DQN}}$.
11:     **every $T$ steps**
12:         Shift the parameters $\bar{\theta}_k \leftarrow \theta_{k+1}$, for $k \in \{0, .., K-1\}$.
13:     **every $D$ steps**
14:         Update $\bar{\theta}_k \leftarrow \theta_k$, for $k \in \{1, .., K-1\}$.

---

**Proposition 4.1.** *Let $t \in \mathbb{N}$, $(\theta_k^t)_{k=0}^K$ be a sequence of parameters of $\Theta$, and $\nu$ be a probability distribution over state-action pairs. If, for every $k \in \{1, .., K\}$,*

$$\underbrace{\|\Gamma^* Q_{\theta_{k-1}^t} - Q_{\theta_k^t}\|_{2,\nu}}_{k^{th} approx\ error\ before\ optimization} - \underbrace{\|\Gamma^* Q_{\theta_{k-1}^t} - Q_{\theta_k^{t+1}}\|_{2,\nu}}_{k^{th} approx\ error\ after\ optimization} \geq \underbrace{\|\Gamma^* Q_{\theta_{k-1}^{t+1}} - \Gamma^* Q_{\theta_{k-1}^t}\|_{2,\nu}}_{displacement\ of\ the\ target} \quad (4)$$

*then, we have*

$$\underbrace{\sum_{k=1}^K \|\Gamma^* Q_{\theta_{k-1}^{t+1}} - Q_{\theta_k^{t+1}}\|_{2,\nu}^2}_{sum\ of\ approx\ errors\ after\ optimization} \leq \underbrace{\sum_{k=1}^K \|\Gamma^* Q_{\theta_{k-1}^t} - Q_{\theta_k^t}\|_{2,\nu}^2}_{sum\ of\ approx\ errors\ before\ optimization} \quad (5)$$

We recall that, when the dataset of samples $\mathcal{D}$ rich enough to represent the true Bellman operator, minimizing $\theta \mapsto \|\Gamma^* Q_{\theta_{k-1}^t} - Q_\theta\|_{2,\nu}^2$ is equivalent to minimizing $\theta \mapsto \sum_{(s,a,r,s') \in \mathcal{D}} \mathcal{L}_{\text{QN}}(\theta | \theta_{k-1}^t, s, a, r, s')$. Crucially, for $t \in \mathbb{N}, k \in \{1, .., K\}$, each $\theta_k^{t+1}$ is learned to minimize $\theta \mapsto \sum_{(s,a,r,s') \in \mathcal{D}} \mathcal{L}_{\text{QN}}(\theta | \theta_{k-1}^t, s, a, r, s')$ starting from $\theta_k^t$. This means that each $\theta_k^{t+1}$ is learned to minimize $\theta \mapsto \|\Gamma^* Q_{\theta_{k-1}^t} - Q_\theta\|_{2,\nu}^2$. Therefore, the optimization procedure is designed to make Equation 4 valid, i.e., to maximize the left-hand side of the inequality. This is why, we argue that Equation 4 is useful to demonstrate that i-QN is theoretically sound. We stress that the presented condition is not meant to be used in practice. Nonetheless, we empirically evaluate the aforementioned quantities in Section 5 to show that the presented condition explains i-QN increased performances.

### 4.1 Practical implementation

The presented approach can be seen as a general framework to design iterated versions of standard value-based algorithms. Indeed, i-QN is an approach orthogonal to the choice of $\mathcal{L}_{QN}$; thus, it can enable multiple simultaneous Bellman updates in any algorithm based on value function estimation. As an example, Algorithm 1 showcases an iterated version of DQN (Mnih et al., 2015) that we call *iterated Deep Q-Network* (i-DQN). We recall that when $K = 1$, the original algorithm is recovered. Similarly, Algorithm 2 is an iterated version of SAC (Haarnoja et al., 2018) that we call *iterated Soft Actor-Critic* (i-SAC). In the actor-critic setting, Polyak averaging (Lillicrap et al., 2015) is usually performed instead of hard updates. Therefore, we adapt the way the targets are updated in i-SAC, i.e., we update the value of $\bar{\theta}_0$ to $\tau\theta_1 + (1-\tau)\bar{\theta}_0$, where $\tau \in [0, 1]$ and the enforced chain structure will shift the rest of the parameters forward.

The availability of a sequence of approximations of consecutive Bellman updates raises the question of how to use them to draw actions in a $Q$-learning setting and how to train the policy in an actor-critic setting. We

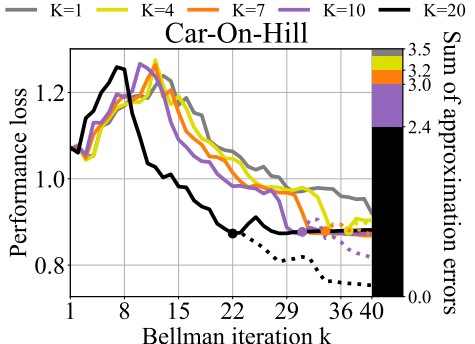

Figure 5: Distance between the optimal value function $V^*$ and $V^{\pi_k}$ at each Bellman iteration $k$ for different $K$.

| $K$ | 1 | 4 | 7 | 10 | 20 |
|---|---|---|---|---|---|
| % of the time the CSAE does not decrease | 0 | 17 | 23 | 23 | 24 |
| Average decrease of the CSAE ($\times 10^{-4}$) | 0.3 | 1.5 | 2.6 | 3.9 | 7.8 |
| % of the time Eq. 5 holds when Eq. 4 holds | 100 | 100 | 100 | 100 | 100 |
| Proportion of the CSAE's decrease when Eq. 4 holds | 100 | 68 | 69 | 66 | 67 |

Table 1: Empirical evaluation of the considered *sum of approximation errors* (CSAE) and Equations 4 and 5 on car-on-hill with i-QN applied to FQI for different window sizes $K$.

consider those choices to be similar as they both relate to the behavioral policy. We recall that, in sequential approaches, the single online network is the only possible choice. For iterated $Q$-Network approaches, we choose to sample a $Q$-function uniformly from the set of online networks to avoid the caveat of passive learning identified by Ostrovski et al. (2021). We empirically justify this choice in Section 6.1.1, where we investigate different sampling strategies.

## 5 Motivating example

In this section, we provide an empirical analysis of i-QN on fitted $Q$-iteration (FQI) in the car-on-hill environment (Ernst et al., 2005) using *non-linear* function approximation (Riedmiller, 2005). The experiment setup is detailed in Appendix C.2. We study i-FQI's behavior with $N = 40$ Bellman iterations for varying window sizes, $K \in \{1, 4, 7, 10, 20\}$. We recall that $K = 1$ corresponds to the regular FQI approach. As the experiment is done offline, we impose a budget on the total number of *non-parallelizable* gradient steps. This means that we have to shift the windows faster for lower values of $K$ so that each algorithm has learned $N = 40$ Bellman updates at the end of the training. On the left y-axis of Figure 5, we show that for higher values of $K$, we obtain lower performance losses (in solid lines) after only a few Bellman iterations. In dashed lines, we show that continuing the training further improves the performance as the iterated approach keeps learning the last $K-1$ projection steps since the considered windows still contain those terms. This motivates the idea of learning consecutive Bellman updates simultaneously.

We now push this analysis further by computing the key quantities introduced in Section 4. On the right y-axis of Figure 5, the sum of the 40 approximation errors decreases as $K$ increases. As expected, this is in accordance with the aforementioned theoretical results of Theorem 3.4 from Farahmand (2011). The effect of performing multiple Bellman updates is also evident from a visual interpretation of the plots, where one can see that increasing the value of $K$ results in shrinking the plots obtained for smaller values of $K$ to the left. This is explained by the ability of i-QN to look ahead of multiple Bellman iterations, thus anticipating the behavior of less far-sighted variants and making better use of the sample. It is important to note that car-on-hill needs approximately 20 FQI iterations to be solved, for which reason i-FQI with $K = 20$ has an evident advantage over the other values of $K$.

As stated in Section 4, i-QN does not minimize the sum of approximation errors directly due to semi-gradient updates. This can be verified empirically from the first line of Table 1 as the percentage of time the considered sum of approximation errors does not decrease (Equation 5 does not hold) is positive when $K > 1$. Nonetheless, it appears that, on average, the considered sum of approximation errors decreases more when $K$ increases, as the second line of Table 1 indicates. This is coherent with the results shown in Figure 5, as the sum of approximation errors is lower for higher values of $K$. The third line of Table 1 demonstrates that when the proposed condition of Proposition 4.1 is verified (Equation 4 holds), the considered sum of approximation errors always decreases (Equation 5 holds). This confirms that this condition is sufficient.

Finally, the last line of the table reports the proportion of the decrease in the considered sum of approximation errors that happens when the condition of Proposition 4.1 is verified (Equation 4 holds) compared to when it is not verified. The proposed condition is responsible for $\approx 70\%$ of the decrease of the considered sum of approximation errors. This shows that the proposed condition is relevant to explain i-QN's ability to minimize the sum of approximation errors for the considered example. For $K = 1$, this metric is at $100\%$, which is expected since the proposed condition is equivalent to a decrease in the considered sum of approximation errors. Indeed, when $K = 1$, the displacement to the target (the left term of Equation 4) is null since $\theta_0^t$, the only target parameter, is constant in $t$.

## 6  Experiments

We evaluate our proposed i-QN approach on deep value-based and actor-critic settings. As recommended by Agarwal et al. (2021), we choose the interquartile mean (IQM) of the human normalized score to report the results of our experiments with shaded regions showing pointwise 95% percentile stratified bootstrap confidence intervals. IQM is a trade-off between the mean and the median where the tail of the score distribution is removed on both sides to consider only 50% of the runs. 5 seeds are used for each Atari game, and 10 seeds are used for each MuJoCo environment.

### 6.1  Atari 2600

We evaluate the iterated version of DQN and implicit quantile network (IQN) (Dabney et al., 2018) on 20 Atari 2600 games. Many implementations of Atari environments along with classical baselines are available (Castro et al., 2018; D'Eramo et al., 2021; Raffin et al., 2021; Huang et al., 2022). We choose to mimic the implementation choices made in Dopamine RL (Castro et al., 2018) since it is the only one to release the evaluation metric for the baselines that we consider and the only one to use the evaluation metric recommended by Machado et al. (2018). Namely, we use the *game over* signal to terminate an episode instead of the *life* signal. The input given to the neural network is a concatenation of 4 frames in grayscale of dimension 84 by 84. To get a new frame, we sample 4 frames from the Gym environment (Brockman et al., 2016) configured with no frame-skip, and we apply a max pooling operation on the 2 last grayscale frames. We use sticky actions to make the environment stochastic (with $p = 0.25$). The performance is the one obtained during training. By choosing an identical setting as Castro et al. (2018), we leverage the baselines' training performance reported in Dopamine RL. To ensure that the comparison is fair, we compared our version of DQN and IQN to their version and verified their equivalence (see Figures 12 in Appendix C.3).

**Hyperparameter settings.** We use the same hyperparameters of the baselines, except for the target update frequency, which we set 25% lower than the target update frequency of the baselines (6000 compared to 8000). This choice is made to benefit from i-QN's ability to learn each projection step with more gradient steps and samples while shifting the window faster than sequential approaches. We stress that i-QN has access to the same number of non-parallelizable gradient steps and the same number of samples. We choose $D = 30$, and we let the $Q$-functions share the convolutional layers. We present the architecture of i-QN in Figure 11 in Appendix C.3. Further details can be found in Table 3 in Appendix C.3. To ensure that our implementation is trustworthy, Figure 12 in Appendix C.3 shows that the training performances of DQN and IQN are comparable to the ones of i-DQN and i-IQN with $K = 1$, as expected. Finally, we note $T$ the target update frequency and $G$ the number of environment interactions per gradient steps.

**Atari results.** i-DQN with $K = 5$ outperforms DQN on the aggregation metric, as shown in Figure 6 (left), both in terms of sample-efficiency and overall performance. In Figure 15 (left) in Appendix E, the performance profile, showing the distribution of final scores, illustrates that i-DQN statistically dominates DQN on most of the scores. Moreover, i-DQN improves over DQN's optimality gap during training as presented in Figure 15 (right). Notably, the iterated version of IQN with $K = 3$ greatly outperforms the sequential approach. All the individual training curves for the 20 Atari games are available in Figure 14 in Appendix E.

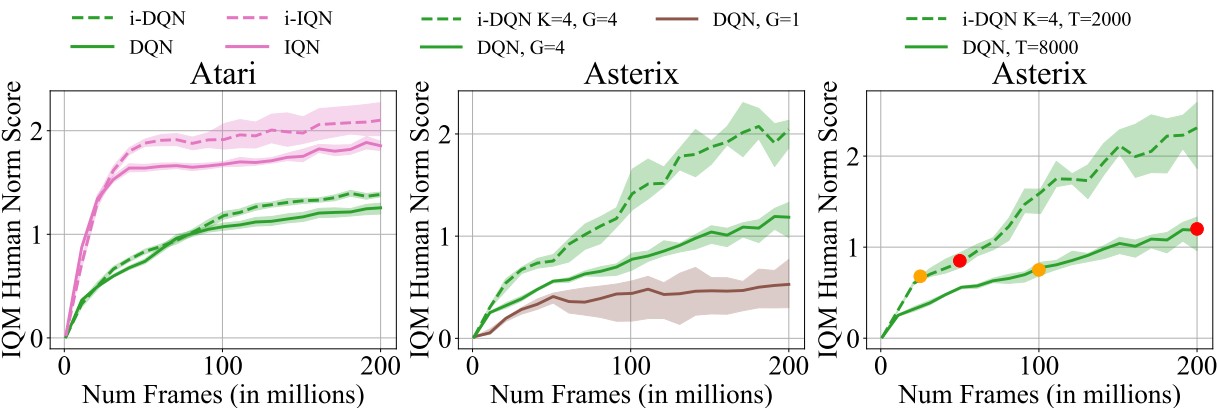

Figure 6: **Left:** i-DQN and i-IQN outperform their respective sequential approach. **Middle:** i-DQN greatly outperforms DQN with $G = 1$, which is a variant of vanilla DQN ($G = 4$) that has access to the same amount of gradient steps as i-DQN. This shows that simply increasing the number of gradient steps of DQN is not effective. **Right:** i-DQN with $K = 4, T = 2000$ parallelizes the training of DQN $T = 8000$ successfully since it yields similar performances after 3125 projection steps (in orange). This parallelization saves samples and gradient steps which can be used later to outperform DQN.

### 6.1.1 Ablation studies

We propose various ablation studies on different Atari games to help understand i-QN's behavior.

**Gradient steps executed in parallel.** The iterated approach performs $K$ times more gradient steps than the sequential one as it considers $K$ online networks instead of 1. Crucially, the batch of samples is shared between the networks, which means that the iterated approach observes the same amount of samples during training. Additionally, those additional gradient steps are executed in parallel so that the training time remains controllable when enough parallel processing power is available. We refer to Appendix D for further discussions. Nonetheless, we compare i-DQN with $K = 4$ (i-DQN, $K = 4, G = 4$) to a version of DQN which has access to the same number of gradient steps (DQN, $G = 1$). We also set the target update frequency of i-DQN $T$ to be the *same* as DQN, i.e., 8000 so that the windows shift at equal speed. We report the performances in Figure 6 (middle) and add a vanilla version of DQN (DQN, $G = 4$) for completeness. DQN with $G = 1$ leads to overfitting while having a training time 89% longer than i-DQN. This result shows that simply allowing DQN to perform more gradient steps is not beneficial.

**Learning consecutive projection steps simultaneously.** In Section 4, we argue that we can effectively learn multiple Bellman updates simultaneously. We also provide a condition under which this idea is guaranteed to be beneficial. We now evaluate a version of i-DQN with $K = 4$, and with a target update frequency set at a fourth of the one of DQN ($T = 2000$ instead of 8000). With this setting, each projection step is learned on the same number of samples and gradient steps with DQN and i-DQN. We report the results in Figure 6 (right) and mark in orange the performance after 3125 ($100 \times 250000/8000 = 25 \times 250000/2000 = 3125$) projection steps and in red the performance after 6250 projection steps. i-DQN requires 4 times fewer samples and gradient steps to reach those marked points compared to DQN since i-DQN's window shifts 4 times faster. Interestingly, the performances of i-DQN and DQN after 3125 projection steps are similar, showing that waiting for one projection step to finish to learn the next one is not necessary. After the 6250 projection steps, the performance of i-DQN is slightly below the one of DQN. We believe that this comes from the fact that, at this point of the training, DQN has access to 4 times more environment interactions than i-DQN. Nevertheless, by learning the 3 following projection steps while the first one is being learned, i-DQN is more efficient than DQN.

**Comparing different window sizes.** In Figure 7 (left), we report the performances for $K = 1, 5$ and 10. Notably, higher performance is achieved with higher window sizes. In the main experiment presented in Figure 6 (left), we choose $K = 5$ because it provides a good trade-off between sample efficiency and additional training time. See Section D for more details about training time.

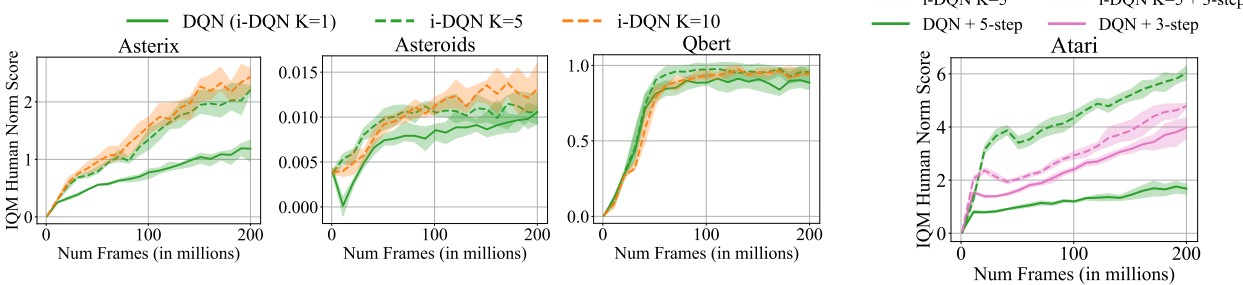

Figure 7: **Left:** Ablation study on the number of Bellman updates $K$ taken into account in the loss. Greater performances are reached for greater values of $K$ in *Asteroids* and *Asterix*. In *Qbert*, iDQN with $K = 10$ might allow the agent to overfit the targets since each Bellman update is learned with 10 times more gradient steps. We recall that DQN is equivalent to iDQN with $K = 1$. **Right:** i-DQN with $K = 5$ performs differently than DQN + 5-step return on 4 randomly selected games (*Breakout, DemonAttack, ChopperCommand*, and *Krull*). DQN + 3-step return can be improved by simultaneously learning 5 Bellman updates.

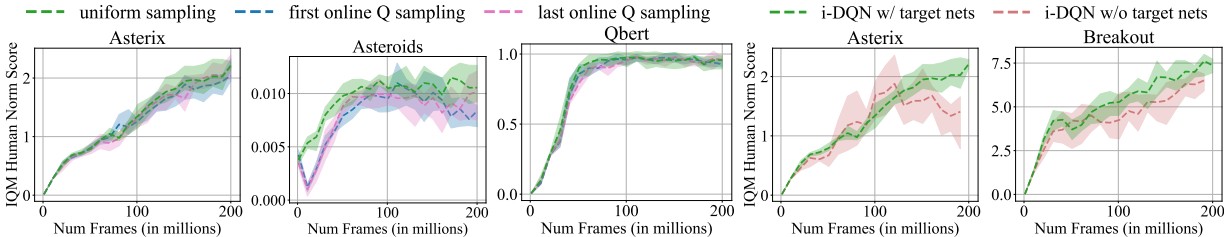

Figure 8: **Left:** Ablation study on how actions are sampled when i-DQN interacts with the environment. **Right:** Ablations on Asterix and Breakout indicate improved performance with target networks.

**Iterated $Q$-network vs. $n$-step return.** We point out that i-QN is not equivalent to $n$-step return (Watkins, 1989). While our approach aims at learning multiple consecutive projection steps simultaneously, $n$-step return applies Bellman updates considering a sequence of $n$ consecutive rewards and the bootstrapped target at the $n^{\text{th}}$ step. In other words, $n$-step return derives another empirical Bellman operator as it computes the target as an interpolation between a one-step bootstrapping and a Monte-Carlo estimate. Nonetheless, we compare i-DQN with $K = 5$ to DQN with 5-step return in Figure 7 (right). DQN with 5-step return behaves substantially worse than i-DQN with $K = 5$ on 4 randomly selected Atari games. The individual training curves are shown in Figure 16 of Appendix E.

One parallel can be done to link the two approaches in the APE setting. Indeed, using $n$-step return can be seen as applying the empirical Bellman operator $n$ times. With $n = 2$, $(\hat{\Gamma}^\pi)^2 Q(s, a) = r_0 + \gamma \hat{\Gamma}^\pi Q(s_1, \pi(s_1))$ $= r_0 + \gamma(r_1 + \gamma Q(s_2, \pi(s_2))) = r_0 + \gamma r_1 + \gamma^2 Q(s_2, \pi(s_2))$. Thus, combining the idea of learning $K$ Bellman updates with $n$-step return artificially brings the length of the window to $n \times K$ where $K$ consecutive projections of the Bellman operator applied $n$ times are learned. This fact is also discussed in Section 5.3 of Schmitt et al. (2022). It is worth noticing that this is not the case for AVI since the max operator is not linear. With $n = 2$, $(\hat{\Gamma}^*)^2 Q(s, a) = r_0 + \gamma \max_{a_1} \hat{\Gamma}^* Q(s_1, a_1) = r_0 + \gamma \max_{a_1}(r_1 + \gamma \max_{a_2} Q(s_2, a_2)) \neq r_0 + \gamma r_1 + \gamma^2 \max_{a_2} Q(s_2, a_2)$. Interestingly, $n$-step return can be seamlessly used with i-QN. In Figure 7, we show that the iterated version of DQN with 3-step return outperforms its sequential counterpart.

**Action sampling.** As explained in Section 4, the availability of several online $Q$-functions corresponding to $K$ consecutive Bellman updates raises the question of which network to use for the sampling actions. Informally, we can argue that along the chain of $Q$-functions in i-DQN, on the one hand, the first network is the one with the best estimate of its respective Bellman update as it has seen the most samples, but it is the most distant from the optimal $Q$-function. On the other hand, the last network is the less accurate but the most far-sighted one. This creates a trade-off between accuracy and distance from the optimal $Q$-function, that we face by sampling a network uniformly at each step. This also helps mitigate passive learning (Ostrovski et al., 2021) as each network can interact with the environment. In Figure 8 (left), we

compare the performance of our uniform sampling strategy to a variant of i-DQN where only the first and last network sample actions. The uniform sampling strategy shows a slight superiority over the others.

**Relevance of the target networks.** We evaluate a version of i-DQN without target networks. In this version, each online parameter $\theta_k$ is learned from the previous online parameter of the chain $\theta_{k-1}$. In Figure 8 (right), we compare this version of i-DQN to the vanilla version of i-DQN where each target parameter $\bar{\theta}_k$ is updated to its respective online parameter $\theta_k$ every $D$ gradient steps. Despite having fewer memory requirements, the version of i-DQN without target networks obtains worse results and even drops performance on *Asterix*. This supports the idea that target networks are useful for stabilizing the training.

In Appendix E, we present additional ablation studies about the choice of sharing the convolutional layers, the distance between the networks during training, and a comparison between i-DQN, a version of DQN with the same total number of parameters as i-DQN, a comparison between i-QN and Bootstrapped DQN (Osband et al., 2016), and we discuss the link between i-QN and Value Iteration Networks (Tamar et al., 2016).

### 6.2 MuJoCo continuous control

We evaluate our proposed iterated approach over two actor-critic algorithms on 6 different MuJoCo environments. We build i-SAC on top of SAC and i-DroQ on top of DroQ (Hiraoka et al., 2022). We set $K = 4$, meaning that the iterated versions consider 4 Bellman updates in the loss instead of 1. Therefore, we set the soft target update rate $\tau$ to 4 times higher than the sequential baseline algorithms ($\tau = 0.02$ compared to 0.005) while keeping all other hyperparameters identical.

Figure 9 shows the training performances of i-SAC and i-DroQ against their respective baselines. The IQM return is normalized by the final performance of SAC. Both iterated versions outperform their respective sequential counterparts. While i-DroQ dominates DroQ until the end of the training at 1 million environment interactions, i-SAC finally converges to the same performances as SAC. Per environment plots are available in Figure 21 of Appendix F. In Figures 20 and 22 of Appendix F, we verify that the gain in performances of i-SAC and i-DroQ over SAC and DroQ is not due to difference in the soft target update rate $\tau$. For that, we make the soft target update rate of SAC and DroQ match the one of i-SAC and i-DroQ. The iterated approaches still outperform the sequential approaches, validating that learning multiple Bellman updates in parallel is beneficial. Table 4 gathers all the hyperparameters used in this section.

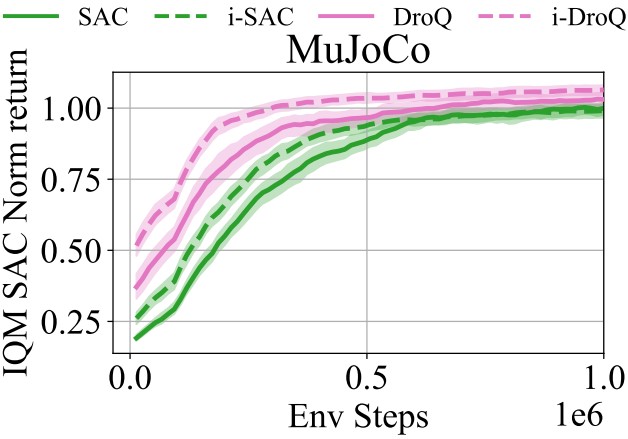

Figure 9: Training curves of i-SAC and i-DroQ ($K = 4$) along with SAC and DroQ. The iterated versions outperform their sequential counterparts. The curves are normalized by the final performance of SAC.

## 7 Discussion and conclusion

We have presented iterated $Q$-Network (i-QN), a new approach that considers multiple consecutive projection steps in the loss to overcome the limitations of the one-step Bellman updates. We have theoretically and empirically analyzed its benefit across several problems when applied to value-based and actor-critic methods. Remarkably, i-IQN provides a 13% improvement over IQN final performances aggregated over 20 Atari games. Future works could investigate the idea of learning several iterations in parallel for other sequential RL algorithms. For example, trust-region methods are sequential algorithms where data collection and policy optimization alternate sequentially.

**Limitations.** While i-QN is sample efficient, it requires additional training time and a larger memory requirement than the sequential approach. We quantify this limitation in Appendix D. Importantly, if enough parallel processing power is available, i-QN's training time becomes similar to the one of a regular $Q$-Network as the additional gradient steps used by i-QN are parallelizable.

**Acknowledgments**

This work was funded by the German Federal Ministry of Education and Research (BMBF) (Project: 01IS22078). This work was also funded by Hessian.ai through the project 'The Third Wave of Artificial Intelligence – 3AI' by the Ministry for Science and Arts of the state of Hessen and by the grant "Einrichtung eines Labors des Deutschen Forschungszentrum für Künstliche Intelligenz (DFKI) an der Technischen Universität Darmstadt".

**Carbon Impact**

As recommended by Lannelongue & Inouye (2023), we used GreenAlgorithms (Lannelongue et al., 2021) and ML $CO_2$ Impact (Lacoste et al., 2019) to compute the carbon emission related to the production of the electricity used for the computations of our experiments. We only consider the energy used to generate the figures presented in this work and ignore the energy used for preliminary studies. The estimations vary between 1.39 and 1.56 tonnes of $CO_2$ equivalent. As a reminder, the Intergovernmental Panel on Climate Change advocates a carbon budget of 2 tonnes of $CO_2$ equivalent per year per person.

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

## A   Proofs

**Theorem 3.4 from** Farahmand (2011). *Let $N \in \mathbb{N}^*$, and $\rho$, $\nu$ two probability measures on $\mathcal{S} \times \mathcal{A}$. For any sequence $(Q_k)_{k=0}^N \in \mathcal{Q}_\Theta^{N+1}$ where $R_\gamma$ depends on the reward function and the discount factor, we have*

$$\|Q^* - Q^{\pi_N}\|_{1,\rho} \leq C_{N,\gamma,R_\gamma} + \inf_{r \in [0,1]} F(r; N, \rho, \gamma) \left( \sum_{k=1}^N \alpha_k^{2r} \|\Gamma^* Q_{k-1} - Q_k\|_{2,\nu}^2 \right)^{\frac{1}{2}}$$

*where $C_{N,\gamma,R_\gamma}$, $F(r; N, \rho, \gamma)$, and $(\alpha_k)_{k=0}^N$ do not depend on $(Q_k)_{k=0}^N$. $\pi_N$ is a greedy policy computed from $Q_N$.*

**Proposition 4.1.** *Let $t \in \mathbb{N}$, $(\theta_k^t)_{k=0}^K$ be a sequence of parameters of $\Theta$, and $\nu$ be a probability distribution over state-action pairs. If, for every $k \in \{1, .., K\}$,*

$$\underbrace{\|\Gamma^* Q_{\theta_{k-1}^t} - Q_{\theta_k^t}\|_{2,\nu}}_{k^{th} approx\ error\ before\ optimization} - \underbrace{\|\Gamma^* Q_{\theta_{k-1}^t} - Q_{\theta_k^{t+1}}\|_{2,\nu}}_{k^{th} approx\ error\ after\ optimization} \geq \underbrace{\|\Gamma^* Q_{\theta_{k-1}^{t+1}} - \Gamma^* Q_{\theta_{k-1}^t}\|_{2,\nu}}_{displacement\ of\ the\ target} \quad (4)$$

*then, we have*

$$\underbrace{\sum_{k=1}^K \|\Gamma^* Q_{\theta_{k-1}^{t+1}} - Q_{\theta_k^{t+1}}\|_{2,\nu}^2}_{sum\ of\ approx\ errors\ after\ optimization} \leq \underbrace{\sum_{k=1}^K \|\Gamma^* Q_{\theta_{k-1}^t} - Q_{\theta_k^t}\|_{2,\nu}^2}_{sum\ of\ approx\ errors\ before\ optimization} \quad (5)$$

*Proof.* For $t \in \mathbb{N}$, let $(\theta_k^t)_{k=0}^K$ be a sequence of parameters of $\Theta$ and $\nu$ be a probability distribution over state-action pairs. We assume that for every $k \in \{1, .., K\}$ condition Equation 4 holds.

Now, we show that for every $k \in \{1, .., K\}$, $\|\Gamma^* Q_{\theta_{k-1}^{t+1}} - Q_{\theta_k^{t+1}}\|_{2,\nu} \leq \|\Gamma^* Q_{\theta_{k-1}^t} - Q_{\theta_k^t}\|_{2,\nu}$. From there, Equation 5 can be obtained by applying the square function to both sides of the inequality and summing over $k$.

Let $k \in \{1, .., K\}$. To show that $\|\Gamma^* Q_{\theta_{k-1}^{t+1}} - Q_{\theta_k^{t+1}}\|_{2,\nu} \leq \|\Gamma^* Q_{\theta_{k-1}^t} - Q_{\theta_k^t}\|_{2,\nu}$, we start with the left side of the inequality

$$\begin{aligned}
\|\Gamma^* Q_{\theta_{k-1}^{t+1}} - Q_{\theta_k^{t+1}}\|_{2,\nu} &= \|\Gamma^* Q_{\theta_{k-1}^{t+1}} - \Gamma^* Q_{\theta_{k-1}^t} + \Gamma^* Q_{\theta_{k-1}^t} - Q_{\theta_k^{t+1}}\|_{2,\nu} \\
&\leq \|\Gamma^* Q_{\theta_{k-1}^{t+1}} - \Gamma^* Q_{\theta_{k-1}^t}\|_{2,\nu} + \|\Gamma^* Q_{\theta_{k-1}^t} - Q_{\theta_k^{t+1}}\|_{2,\nu} \\
&\leq \|\Gamma^* Q_{\theta_{k-1}^t} - Q_{\theta_k^t}\|_{2,\nu},
\end{aligned}$$

The second last inequation comes from the triangular inequality, and the last inequation comes from the assumption that Equation 4 holds. □

**Proposition A.1.** *Let $(\bar{\theta}, \theta)$ be a pair of parameters of $\Theta$ and $\mathcal{D} = \{(s, a, r, s')\}$ be a set of samples. Let $\nu$ be the distribution represented by the state-action pairs present in $\mathcal{D}$. We note $\mathcal{D}_{s,a} = \{(r, s')|(s, a, r, s') \in \mathcal{D}\}, \forall (s, a) \in \mathcal{D}$. If for every state-action pair $(s, a) \in \mathcal{D}$, $\mathbb{E}_{(r,s') \sim \mathcal{D}_{s,a}} \left[ \hat{\Gamma} Q_{\bar{\theta}}(s, a) \right] = \Gamma Q_{\bar{\theta}}(s, a)$, then,*

$$\sum_{(s,a,r,s') \in \mathcal{D}} \mathcal{L}_{QN}(\theta | \bar{\theta}, s, a, r, s') = M \|\Gamma^* Q_{\bar{\theta}} - Q_\theta\|_{2,\nu}^2 + constant\ w.r.t.\ \theta \quad (6)$$

*where $M$ is a constant w.r.t. $\theta$. Thus, minimizing $\theta \mapsto \sum_{(s,a,r,s') \in \mathcal{D}} \mathcal{L}_{QN}(\theta | \bar{\theta}, s, a, r, s')$ is equivalent to minimizing $\theta \mapsto \|\Gamma^* Q_{\bar{\theta}} - Q_\theta\|_{2,\nu}^2$.*

*Proof.* The proof is inspired from Vincent et al. (2025). Let $(\bar{\theta}, \theta)$ be a pair of parameters of $\Theta$ and $\mathcal{D} = \{(s, a, r, s')\}$ be a set of samples. Let $\nu$ be the distribution of the state-action pairs present in $\mathcal{D}$. In this proof, we note $\hat{\Gamma}$ as $\hat{\Gamma}_{r,s'}$ to stress its dependency on the reward $r$ and the next state $s'$. For every state-action pair $(s, a)$ in $\mathcal{D}$, we define the set $\mathcal{D}_{s,a} = \{(r, s') | (s, a, r, s') \in \mathcal{D}\}$ and assume that $\mathbb{E}_{(r,s') \sim \mathcal{D}_{s,a}}[\hat{\Gamma}_{r,s'} Q_{\bar{\theta}}(s, a)] = \Gamma Q_{\bar{\theta}}(s, a)$. Additionally, we note $M$ the cardinality of $\mathcal{D}$, $M_{s,a}$ the cardinality of $\mathcal{D}_{s,a}$ and $\mathring{\mathcal{D}}$ the set of unique state-action pairs in $\mathcal{D}$. We write

$$
\sum_{(s,a,r,s') \in \mathcal{D}} \mathcal{L}_{\mathrm{QN}}(\theta | \bar{\theta}, s, a, r, s') = \sum_{(s,a,r,s') \in \mathcal{D}} \left( \hat{\Gamma}_{r,s'} Q_{\bar{\theta}}(s, a) - Q_{\theta}(s, a) \right)^2
$$

$$
= \sum_{(s,a,r,s') \in \mathcal{D}} \left( \hat{\Gamma}_{r,s'} Q_{\bar{\theta}}(s, a) - \Gamma Q_{\bar{\theta}}(s, a) + \Gamma Q_{\bar{\theta}}(s, a) - Q_{\theta}(s, a) \right)^2
$$

$$
= \sum_{(s,a,r,s') \in \mathcal{D}} \left( \hat{\Gamma}_{r,s'} Q_{\bar{\theta}}(s, a) - \Gamma Q_{\bar{\theta}}(s, a) \right)^2
$$

$$
+ \sum_{(s,a,r,s') \in \mathcal{D}} \left( \Gamma Q_{\bar{\theta}}(s, a) - Q_{\bar{\theta}}(s, a) \right)^2
$$

$$
+ 2 \sum_{(s,a,r,s') \in \mathcal{D}} \left( \hat{\Gamma}_{r,s'} Q_{\bar{\theta}}(s, a) - \Gamma Q_{\bar{\theta}}(s, a) \right) \left( \Gamma Q_{\bar{\theta}}(s, a) - Q_{\theta}(s, a) \right).
$$

The second last equation is obtained by introducing the term $\Gamma Q_{\bar{\theta}}(s, a)$ and removing it. The last equation is obtained by developing the previous squared term. Now, we study each of the three terms:

- $\sum_{(s,a,r,s') \in \mathcal{D}} \left( \hat{\Gamma}_{r,s'} Q_{\bar{\theta}}(s, a) - \Gamma Q_{\bar{\theta}}(s, a) \right)^2$ is independent of $\theta$

- $\sum_{(s,a,r,s') \in \mathcal{D}} \left( \Gamma Q_{\bar{\theta}}(s, a) - Q_{\theta}(s, a) \right)^2$ equal to $M \times ||\Gamma Q_{\bar{\theta}} - Q_{\theta}||_{2,\nu}^2$ by definition of $\nu$.

-
$$
\sum_{(s,a,r,s') \in \mathcal{D}} \left( \hat{\Gamma}_{r,s'} Q_{\bar{\theta}}(s, a) - \Gamma Q_{\bar{\theta}}(s, a) \right) \left( \Gamma Q_{\bar{\theta}}(s, a) - Q_{\theta}(s, a) \right)
$$

$$
= \sum_{(s,a) \in \mathring{\mathcal{D}}} \left[ \sum_{(r,s') \in \mathcal{D}_{s,a}} \left( \hat{\Gamma}_{r,s'} Q_{\bar{\theta}}(s, a) - \Gamma Q_{\bar{\theta}}(s, a) \right) \left( \Gamma Q_{\bar{\theta}}(s, a) - Q_{\theta}(s, a) \right) \right] = 0
$$

since, for every $(s, a) \in \mathring{\mathcal{D}}$,

$$
\sum_{(r,s') \in \mathcal{D}_{s,a}} \left( \hat{\Gamma}_{r,s'} Q_{\bar{\theta}}(s, a) - \Gamma Q_{\bar{\theta}}(s, a) \right) = M_{s,a} \left( \mathbb{E}_{(r,s') \sim \mathcal{D}_{s,a}}[\hat{\Gamma}_{r,s'} Q_{\bar{\theta}}(s, a)] - \Gamma Q_{\bar{\theta}}(s, a) \right) = 0,
$$

the last equality holds from the assumption.

Thus, we have

$$
\sum_{(s,a,r,s') \in \mathcal{D}} \mathcal{L}_{\mathrm{QN}}(\theta | \bar{\theta}, s, a, r, s') = M \times ||\Gamma Q_{\bar{\theta}} - Q_{\theta}||_{2,\nu}^2 + \text{constant w.r.t } \theta
$$

This is why minimizing $\theta \mapsto \sum_{(s,a,r,s') \in \mathcal{D}} \mathcal{L}_{\mathrm{QN}}(\theta | \bar{\theta}, s, a, r, s')$ is equivalent to minimizing $\theta \mapsto ||\Gamma^* Q_{\bar{\theta}} - Q_{\theta}||_{2,\nu}^2$. □

## B Pseudocodes

---

**Algorithm 2** Iterated Soft Actor-Critic (i-SAC). Modifications to SAC are marked in purple.

---

1: Initialize the policy parameters $\phi$, $2 \times (K+1)$ parameters $((\theta_k^1, \theta_k^2))_{k=0}^K$, and an empty replay buffer $\mathcal{D}$.
2: **repeat**
3:     Take action $a_t \sim \pi_\phi(\cdot|s_t)$; Observe reward $r_t$, next state $s_{t+1}$; $\mathcal{D} \leftarrow \mathcal{D} \bigcup \{(s_t, a_t, r_t, s_{t+1})\}$.
4:     **for** UTD updates **do**
5:         Sample a mini-batch $\mathcal{B} = \{(s, a, r, s')\}$ from $\mathcal{D}$.
6:         **for** $k = 1, .., K$; $i = 1, 2$ **do** [*in parallel*]
7:         Compute the loss w.r.t. $\theta_k^i$,

$$\mathcal{L}_{\text{SAC}} = \sum_{(s,a,r,s') \in \mathcal{B}} \left( r + \gamma \left( \min_{j \in \{1,2\}} Q_{\theta_{k-1}^j}(s', a') - \alpha \log \pi_\phi(a'|s') \right) - Q_{\theta_k^i} \right)^2, \text{ where } a' \sim \pi_\phi(\cdot|s').$$

8:         Update $\theta_k^i$ from $\nabla_{\theta_k^i} \mathcal{L}_{\text{SAC}}$.
9:         Update $\theta_0^i \leftarrow \tau \theta_1^i + (1 - \tau)\theta_0^i$, for $i \in \{1, 2\}$.
10:         Sample $k^b \sim U\{1, .., K\}$.
11:         Compute the policy loss w.r.t $\phi$, $\mathcal{L}_{\text{Actor}} = \min_{i \in \{1,2\}} Q_{\theta_{k^b}^i}(s, a) - \alpha \log \pi_\phi(a|s)$, where $a \sim \pi_\phi(\cdot|s)$.
12:     Update $\phi$ from $\nabla_\phi \mathcal{L}_{\text{Actor}}$.

---

## C Experiments details

We used the optimizer Adam (Kingma & Ba, 2015) for all experiments. We used JAX (Bradbury et al., 2018) as a deep learning framework. The neural networks used by the i-QN approaches in the experiments presented in Section 6 are initialized following the same procedure as the neural networks used for their respective sequential counterparts.

### C.1 Behavior of $Q$-network and iterated $Q$-network on a Linear Quadratic Regulator

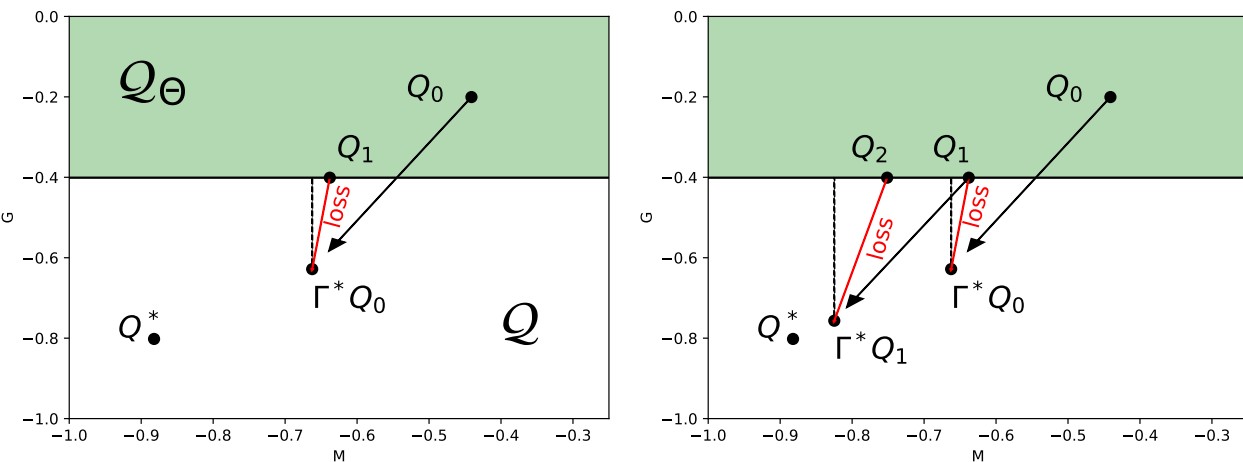

Figure 10: Graphical representation of QN (left) and i-QN (right) in the space of $Q$-functions $\mathcal{Q}$ for the LQR experiment.

Figures 2a and 2b are schematically representing the behavior of QN and i-QN approaches in the space of $Q$-functions. Figure 10 shows that those representations are accurate for an offline problem: Linear Quadratic Regulator (Bradtke, 1992). In this problem, the state and action spaces are continuous and one-dimensional. The dynamics are linear: for a state $s$ and an action $a$, the next state is given by $s' = 0.8s - 0.9a$, and the reward is quadratic $r(s, a) = 0.5s^2 + 0.4sa - 0.5a^2$. We choose to parametrize the space of $Q$-functions with

2 parameters $(M, G)$ such that, for a state $s$ and an action $a$, $Q(s, a) = Ma^2 + Gs^2$. To reduce the space of representable $Q$-functions, we constrain the parameter $M$ to be negative and the parameter $G$ to be between $-0.4$ and $0.4$. Starting from some initial parameters, we perform 30 gradient steps with a learning rate of $0.05$ using the loss of QN and i-QN. Both figures show the space of representable $Q$-functions $\mathcal{Q}_\Theta$ in green, the optimal $Q$-function $Q^*$, the initial $Q$-function $Q_0$ and its optimal Bellman update $\Gamma^* Q_0$. The projection of the optimal Bellman update is also shown with a dotted line. As we claim in the main paper, i-QN manages to find a $Q$-function $Q_2$ closer to the optimal $Q$-function $Q^*$ than $Q_1$ found by QN. Figure 10 (left) closely resembles Figure 2a. Likewise, Figure 10 (right) looks like Figure 2b, showing that the high-level ideas presented in the paper are actually happening in practice.

## C.2 Experiments on the car-on-hill environment

**Experimental setting.** The dataset contains 50.000 samples collected with a uniform policy from the initial state $[-0.5, 0]$. We use a neural network with one hidden layer of 50 neurons. The batch size is 100 and we set $D = 1$. Each i-FQI run has access to 20.000 gradient steps. The results are average over 20 seeds. In Figure 5, we report the performance loss $\|Q^* - Q^{\pi_N}\|_{\rho,1}$, where $Q^{\pi_N}$ is the action-value function associated with the greedy policy of $Q_N$, which can be associated with the last network learned during training. $\rho$ is usually associated to the state-action distribution on which we would like the chosen policy to perform well. Therefore, we choose $\rho$ as a uniform distribution over the state space and action space on a $17 \times 17$ grid as suggested by Ernst et al. (2005). In Figure 5, we compute the sum of approximation errors $\sum_{k=1}^{40} \|\Gamma^* Q_{k-1} - Q_k\|_{\nu,2}^2$, where $\nu$ is the state-action distribution encountered during training, i.e., the data stored in the replay buffer.

## C.3 Experiments on the Atari games

**Atari game selection.** The Atari benchmark is a highly compute-intensive benchmark. Depending on the hardware and the codebase, one seed for one Atari game can take between 1 day to 3 days to run DQN on a GPU. This is why we could not afford to run the experiments on the 57 games with 5 seeds. The 20 games were chosen before doing the experiments and never changed afterward. They were chosen such that the baselines and Rainbow have almost the same aggregated final scores for the 20 chosen games as the aggregated final scores shared by Dopamine RL as shown in Table 2. The Atari games used for the ablation studies were randomly selected. We stress that the purpose of those ablations is to highlight i-QN's behavior in the presented games, it is not meant to draw some conclusions on the general performance of i-QN over the entire benchmark.

Table 2: The IQM Human normalized final scores of the baselines and Rainbow aggregated over the 55 available Atari games are similar to the ones aggregated over the 20 chosen games.

|  | DQN | Rainbow | IQN |
|---|---|---|---|
| Final score over the 55 Atari games available in Dopamine RL | 1.29 | 1.71 | 1.76 |
| Final score over the 20 chosen Atari games | 1.29 | 1.72 | 1.85 |

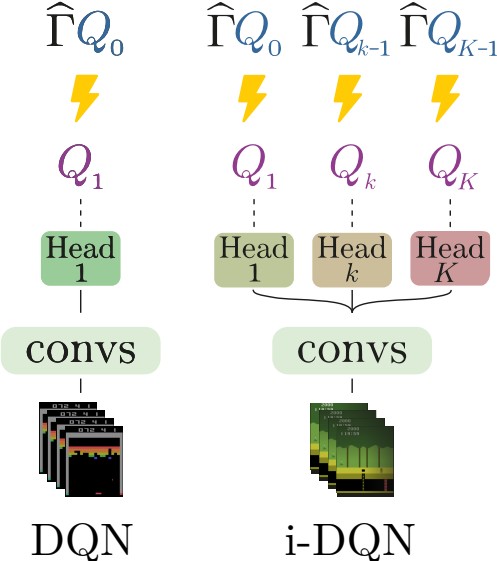

Figure 11: Losses and neural networks architectures of DQN and i-DQN. The dotted lines link the outputs of the neural networks to the objects they represent. The flash signs stress how each projection step is being learned, where $\hat{\Gamma}$ is the empirical Bellman operator. The target networks are represented in blue while the online networks are in purple. For i-DQN, the convolution layers of $Q_0$ are stored separately since it is the only target $Q$-function that is not updated every $D$ steps.

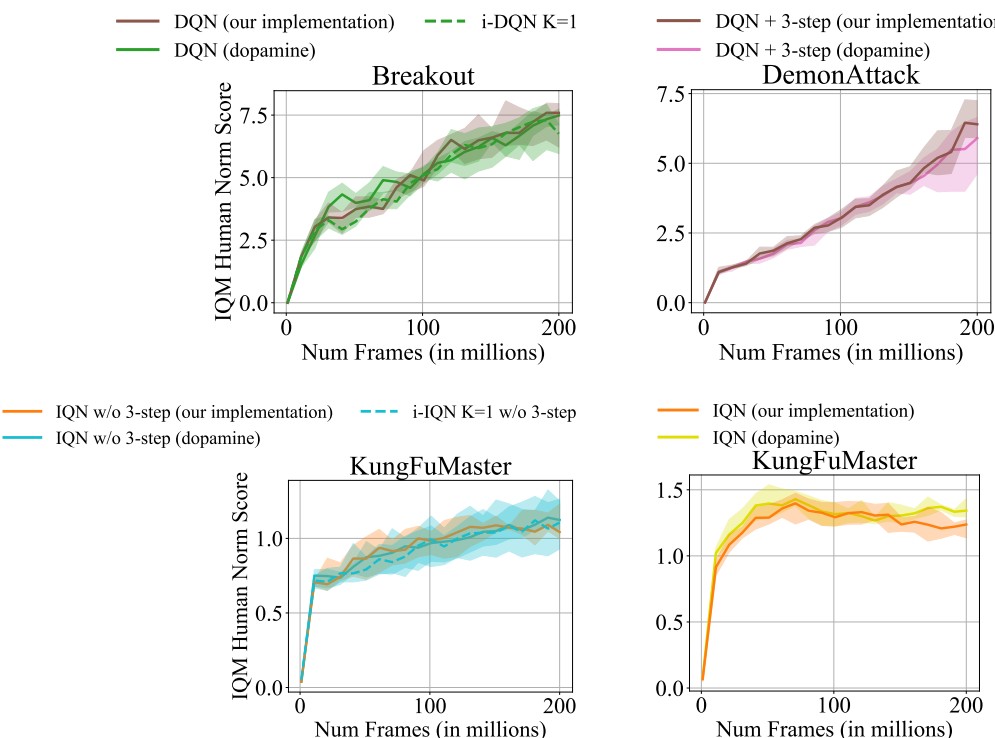

Figure 12: **Left:** Our implementations of DQN (top) and IQN (bottom) yield similar performance as the implementation of Dopamine RL. This certifies that we can compare the results released in Dopamine RL with our method. At the top, DQN and i-DQN with $K = 1$ have a similar behavior. This certifies the trustworthiness of our code base. The same applies to IQN and i-IQN with $K = 1$ at the bottom. **Right:** We draw similar conclusions when adding a 3-step return.

Table 3: Summary of all hyperparameters used for the Atari experiments. We note $\mathrm{Conv}_{a,b}^{d}C$ a 2D convolutional layer with $C$ filters of size $a \times b$ and of stride $d$, and FC $E$ a fully connected layer with $E$ neurons.

| Environment | |
|---|---|
| Discount factor $\gamma$ | 0.99 |
| Horizon $H$ | 27 000 |
| Full action space | No |
| Reward clipping | $\mathrm{clip}(-1, 1)$ |
| All algorithms | |
| Number of epochs | 200 |
| Number of training steps per epochs | 250 000 |
| Type of the replay buffer $\mathcal{D}$ | FIFO |
| Initial number of samples in $\mathcal{D}$ | 20 000 |
| Maximum number of samples in $\mathcal{D}$ | 1 000 000 |
| Gradient step frequency $G$ | 4 |
| Starting $\epsilon$ | 1 |
| Ending $\epsilon$ | $10^{-2}$ |
| $\epsilon$ linear decay duration | 250 000 |
| Batch size | 32 |
| Learning rate | $6.25 \times 10^{-5}$ |
| Adam $\epsilon$ | $1.5 \times 10^{-4}$ |
| Torso architecture | $\mathrm{Conv}_{8,8}^{4}32$ $-\mathrm{Conv}_{4,4}^{2}64$ $-\mathrm{Conv}_{3,3}^{1}64$ |
| Head architecture | $-\mathrm{FC}\ 512$ $-\mathrm{FC}\ n_{\mathcal{A}}$ |
| Activations | ReLU |
| DQN & IQN | |
| Target update frequency $T$ | 8 000 |
| i-DQN & i-IQN | |
| Target update frequency $T$ | 6 000 |
| $D$ | 30 |

Table 4: Summary of all hyperparameters used for the MuJoCo experiments. We note FC $E$ a fully connected layer with $E$ neurons.

| Environment | |
|---|---|
| Discount factor $\gamma$ | 0.99 |
| Horizon $H$ | 1 000 |
| All algorithms | |
| Number of training steps | 1 000 000 |
| Type of the replay buffer $\mathcal{D}$ | FIFO |
| Initial number of samples in $\mathcal{D}$ | 5 000 |
| Maximum number of samples in $\mathcal{D}$ | 1 000 000 |
| Update-To-Data UTD | 1 for SAC/i-SAC 20 for DroQ/i-DroQ |
| Batch size | 256 |
| Learning rate | $10^{-3}$ |
| Adam $\beta_1$ | 0.9 |
| Policy delay | 1 |
| Actor and critic architecture | FC 256 $-\mathrm{FC}\ 256$ |
| SAC & DroQ | |
| Soft target update frequency $\tau$ | $5 \times 10^{-3}$ |
| i-SAC & i-DroQ | |
| Soft target update frequency $\tau$ | $2 \times 10^{-2}$ |
| $D$ | 1 |

# D Training time and memory requirements

In Figure 13, we report the performances presented in the main experiments of the paper (Figure 6 (left) and Figure 9) with the training time as the $x$-axis instead of the number of environment interactions. Computations are made on an NVIDIA GeForce RTX 4090 Ti. For iDQN and iIQN, the game *Breakout* is used to compute the training time. For iSAC and iDroQ, the training time is averaged over the 6 considered environments. The iterated approach requires longer training time compared to the sequential approach. Notably, i-QN reaches the final performances of the sequential approach faster as the first line of Table 5 reports. Indeed, i-IQN reaches IQN's final performance 18h23m before IQN finished its training. Nonetheless, we stress that the main focus of this work is on sample efficiency, as it remains the main bottleneck in practical scenarios. We recall that DroQ and i-DroQ use a UTD of 20, which slows down the training significantly. We also report the additional GPU vRAM usage of i-QN compared to QN on the second line of Table 5. The additional memory requirements remain reasonable. Finally, while the number of additional FLOPs increases linearly with $K$ for a gradients update, sampling an action from an i-QN agent does not take more FLOPs than sampling an action from a QN agent as each $Q$-network has the same architecture (see Table 5).

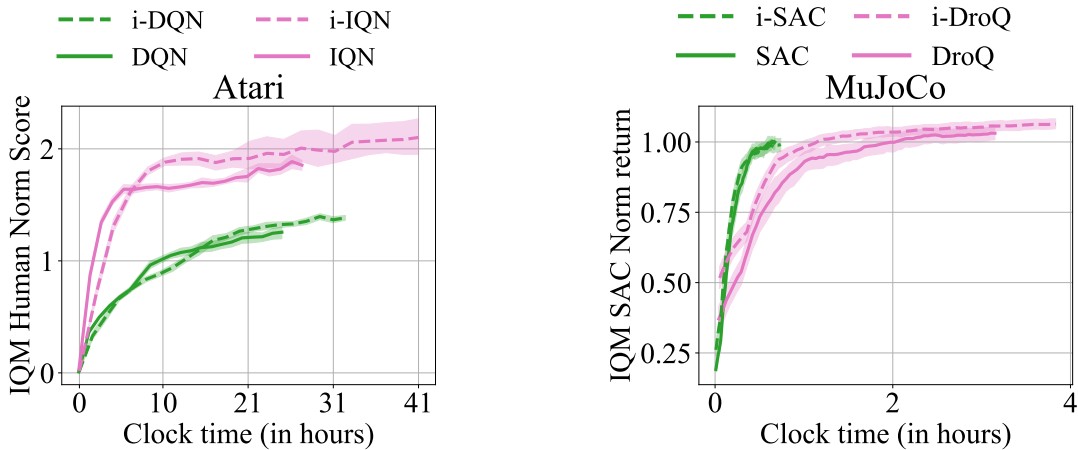

Figure 13: i-QN and QN performances according to the training time on 20 Atari games (left) and 6 MuJoCo environments (right). i-QN requires more training time than QN approaches. Nonetheless, i-QN reaches QN's final performances faster.

Table 5: **First line:** Difference between the time i-QN took to reach QN's final performance and QN's training time as a measure of time efficiency. Despite taking longer to train, i-QN is always more time-efficient than QN's approaches for the four considered instances. **Second line:** The additional GPU vRAM usage of i-QN remains reasonable. **Third line:** The additional FLOPs for a gradient update scale with $K$. **Fourth line:** There are no additional FLOPs for sampling an action as each $Q$-network has the same architecture.

| | i-DQN vs DQN for $K = 5$ | i-IQN vs IQN for $K = 3$ | i-SAC vs SAC for $K = 4$ | i-DroQ vs DroQ for $K = 4$ |
|---|---|---|---|---|
| Time saved by i-QN to reach QN's final performance | 7h31m | 18h23m | 6m | 1h55m |
| Additional GPU vRAM usage | +0.3 Gb | +0.9 Gb | +0.1 Gb | +0.4 Gb |
| Additional FLOPs for a gradient update on a batch of samples | ×5.6 | ×2.2 | ×4.4 | ×4.6 |
| Additional FLOPs for sampling an action | ×1 | ×1 | ×1 | ×1 |

# E    Training curves for Atari games

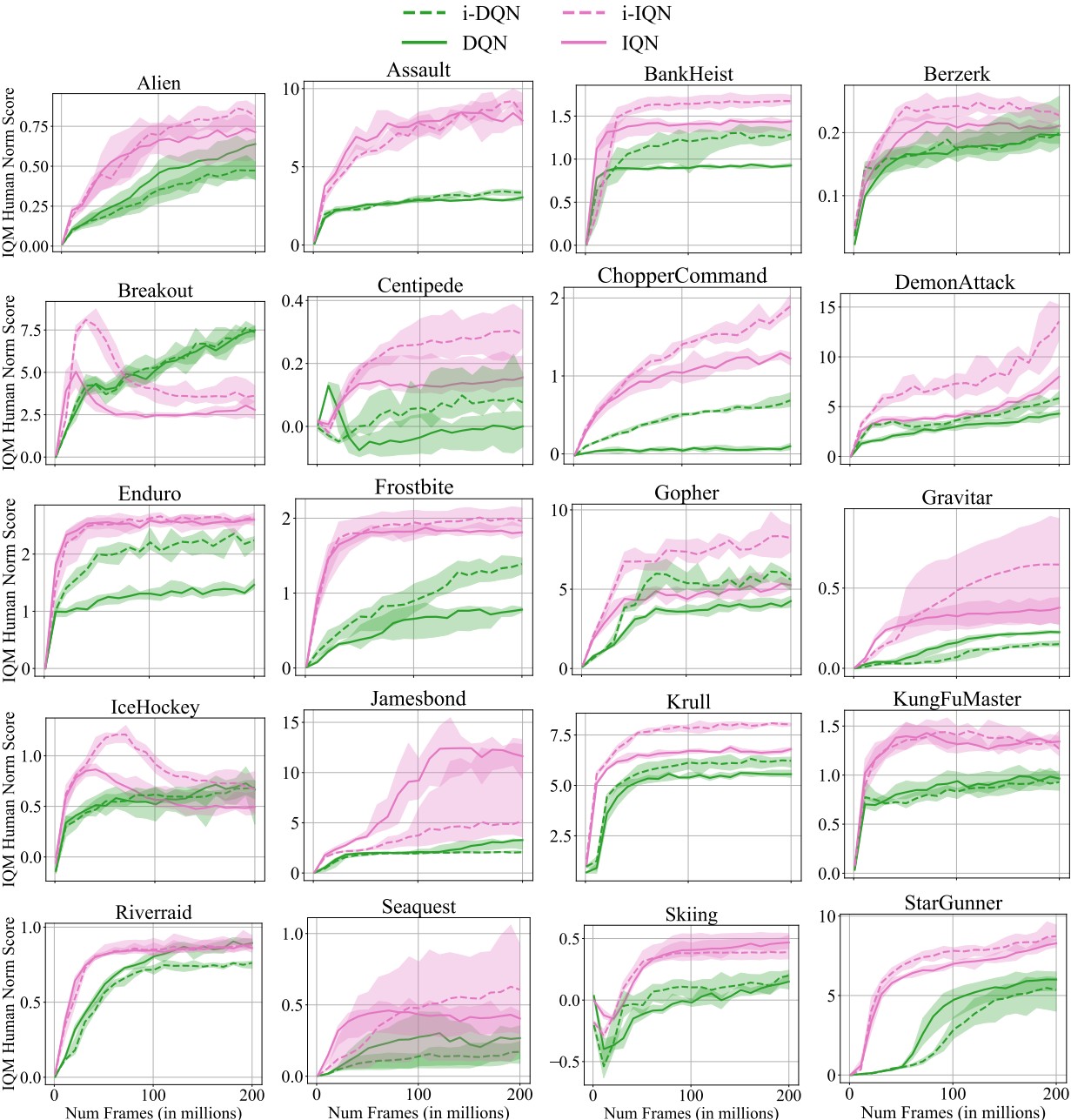

Figure 14: Training curves of i-DQN with $K = 5$ and i-IQN with $K = 3$ along with DQN, Rainbow and i-IQN on 20 Atari games. As a reminder, IQN is also using a 3-step return. In most games, the iterated approach outperforms its respective sequential approach.

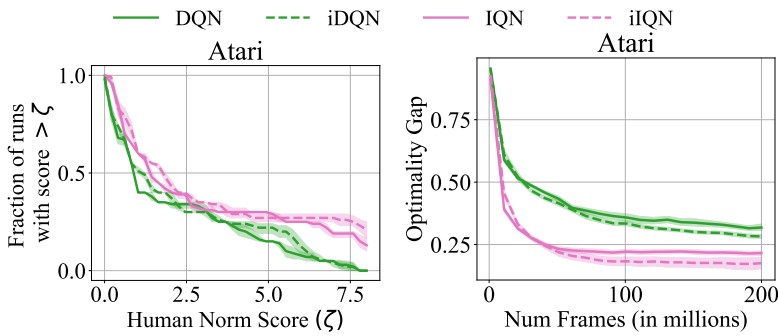

Figure 15: **Left:** Performance profile. The figure shows the fraction of runs with a higher final score than a certain threshold given by the $x$-axis. The iterated versions, i-DQN and i-IQN, statistically dominate their respective sequential approach on most of the domain. **Right:** Optimality gap. The figure shows the average distance to the human level of the runs that do not reach super-human performances. i-DQN and i-IQN achieve lower optimality gap, meaning that those algorithms also improve on games where human performance is not reached by their sequential counterparts.

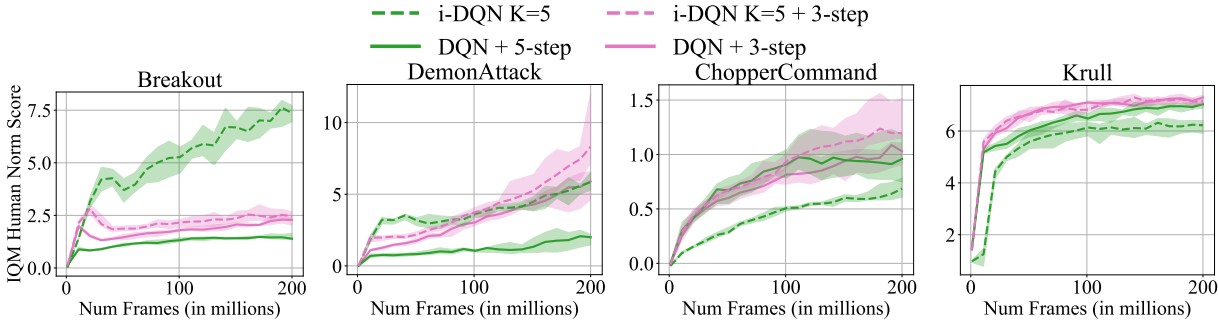

Figure 16: i-DQN with $K = 5$ behaves differently than DQN + 5-step return. DQN + 3-step return is boosted when learning $K = 5$ Bellman updates simultaneously (i-DQN $K = 5$ + 3-step).

**Shared architecture vs. Independent networks.** In Figure 17, we show the performance of i-DQN when the convolution layers are shared and when the neural networks are independent from each other on 2 Atari games. In *ChopperCommand*, having fully independent networks seems more beneficial than sharing the convolutional layers. We conjecture that this is due to the fact that the Bellman updates are far away from each other, hence the difficulty of representing consecutive Bellman updates with shared convolutional

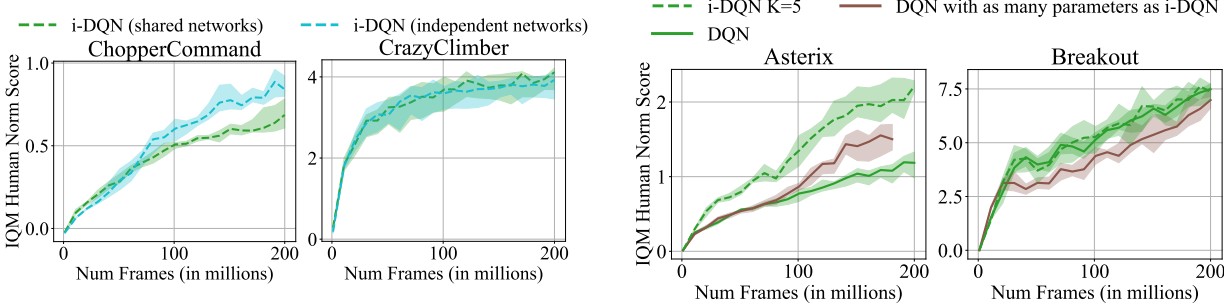

Figure 17: **Left:** Independent networks can perform better than networks with shared convolutions. **Right:** Inflating the number of parameters of DQN until it reaches the same number of parameters i-DQN with $K = 5$ uses does not lead to performances as high as i-DQN on the 2 considered games. Notably, while i-DQN uses the same architecture as DQN, the inflated version of DQN uses a larger architecture.

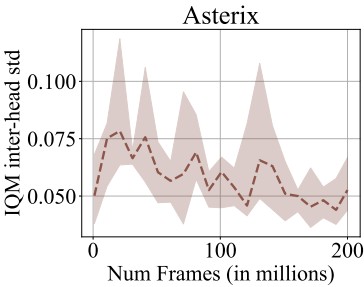 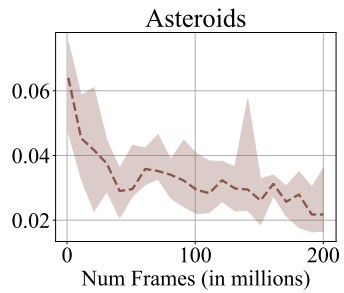 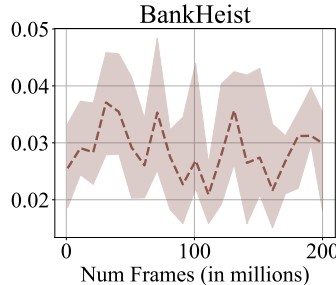

Figure 18: Standard deviation of the output of the 5 online networks of i-DQN averaged over 3200 samples. The standard deviation is greater than zero, indicating that the online networks are different from each other. The signal has a tendency to decrease, which matches our intuition that the $Q$-functions become increasingly close to each other as they get closer to the optimal $Q$-function.

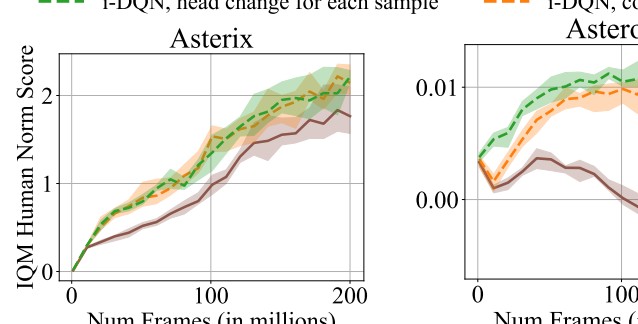 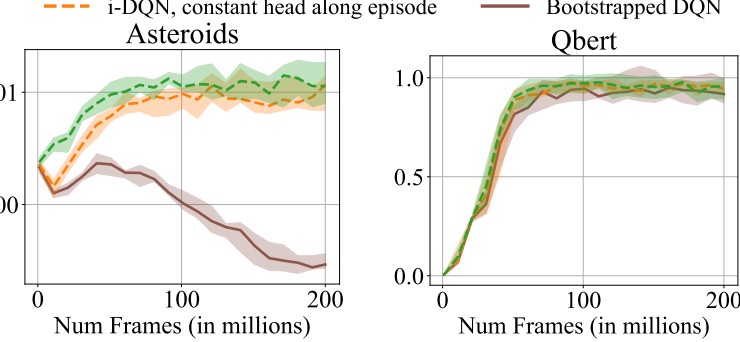

Figure 19: i-DQN (i-DQN, head change for each sample) outperforms Bootstrapped DQN on the 3 considered Atari games. Keeping a fixed head along each episode (i-DQN, constant head along episode) does not seem beneficial compared to sampling a new head at each environment interaction.

layers. Agarwal et al. (2020) share the same conclusion for Random Ensemble Mixture (REM) (Agarwal et al., 2020). They explain that independent networks are more likely to cover a wider space in the space of $Q$-functions. In *CrazyClimber*, both algorithms converge to the same score. Nonetheless, we argue that a shared architecture provides a reasonable trade-off between increased performances and additional memory computation and training time.

**Distance between the online networks.** i-DQN heavily relies on the fact that the learned Q-functions are located within different areas in the space of $Q$-functions. To verify this assumption, we computed the standard deviation of the output of the learned $Q$-functions during the training in Figure 18. This figure shows that the standard deviation among the $Q$-functions is indeed greater than zero across the 3 studied games. Furthermore, the standard deviation decreases during training, suggesting they become increasingly closer. This matches the intuition that at the end of the training, the iteration of the $Q$-functions should lie at the boundary of the space of the space of representable $Q$-functions, close to each other.

**i-DQN vs. DQN with an inflated number of parameters.** i-DQN requires more parameters than DQN as it learns $K$ Bellman updates in parallel. We point out that each network of i-DQN uses the same number of parameters as DQN, which means that both algorithms use the same amount of resources at inference time. Nevertheless, we consider a version of DQN having access to the same number of parameters as i-DQN by increasing the number of neurons in the last hidden layer. In Figure 17, the inflated version of DQN does not provide an increase in performance as high as i-DQN. This supports the idea that i-QN's benefit lies in how the networks are organized to minimize the sum of approximation errors, as explained in Section 4.

**i-QN vs. Bootstrapped DQN (Osband et al., 2016).** Bootstrapped DQN is a method that also uses an ensemble of $N$ $Q$-functions. However, Bootstrapped DQN and i-QN have orthogonal objectives: while Bootstrapped DQN focuses on deep exploration, i-QN aims at learning multiple Bellman updates concurrently. Nevertheless, in Figure 19, we propose to compare Bootstrapped DQN with $N = 6$ to i-DQN with $K = 5$ (i-DQN, head change for each sample) such that both algorithms use the same number of $Q$-functions. Indeed, i-DQN requires $K + 1$ networks. i-DQN outperforms Bootstrapped DQN on the 3 considered Atari games. Additionally, we evaluate a version of i-DQN where the same $Q$-function is kept for interacting with the environment (i-DQN, constant head along episode). This version of i-DQN performs slightly worse that the version of i-DQN where a head is sampled at each environment interaction. This suggests that keeping the same head along an episode to hope for deeper exploration, as suggested by Osband et al. (2016), is less efficient than sampling a head at each environment interaction to avoid learning the other heads passively as identified by Ostrovski et al. (2021).

**i-QN vs. Value Iteration Networks (VIN) (Tamar et al., 2016).** While i-QN's goal is to learn multiple Bellman updates concurrently, the goal of VINs is to train a policy end-to-end and to include planning steps in a finite MDP $\bar{\mathcal{M}}$ that is different from the original MDP $\mathcal{M}$ inside the policy architecture. This means that while i-QN's objective is to learn multiple Q-functions at once, VIN's objective is to learn the transition dynamics and the reward of the finite MDP $\bar{\mathcal{M}}$. This is why, the VIN framework entails some drawbacks that do not belong to the i-QN framework making the comparison impractical. Indeed, VINs do not scale well with the number of Bellman iterations (planning steps) needed for solving the task because the number of parameters used by VINs scales linearly with the number of planning steps as opposed to i-QN where the considered window of Bellman updates can be effectively shifted as explained in Section 4. Additionally, as opposed to i-QN, VINs assume that, for each MDP $\mathcal{M}$, there exists a \*finite\* MDP $\bar{\mathcal{M}}$ such that the optimal plan for $\bar{\mathcal{M}}$ contains useful information about the optimal policy of the original MDP $\mathcal{M}$. This assumption is not guaranteed to hold for all MDPs and more importantly, some design choices need to be made to model $\bar{\mathcal{M}}$ in practice.

# F   Training curves for MuJoCo control problems

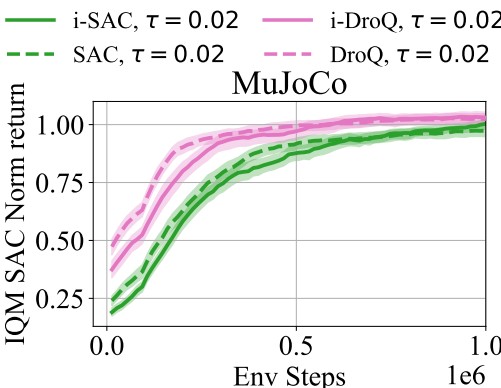

Figure 20: The iterated versions of SAC and DroQ with $K = 4$ also outperform their sequential versions when all the hyperparameters are the same.

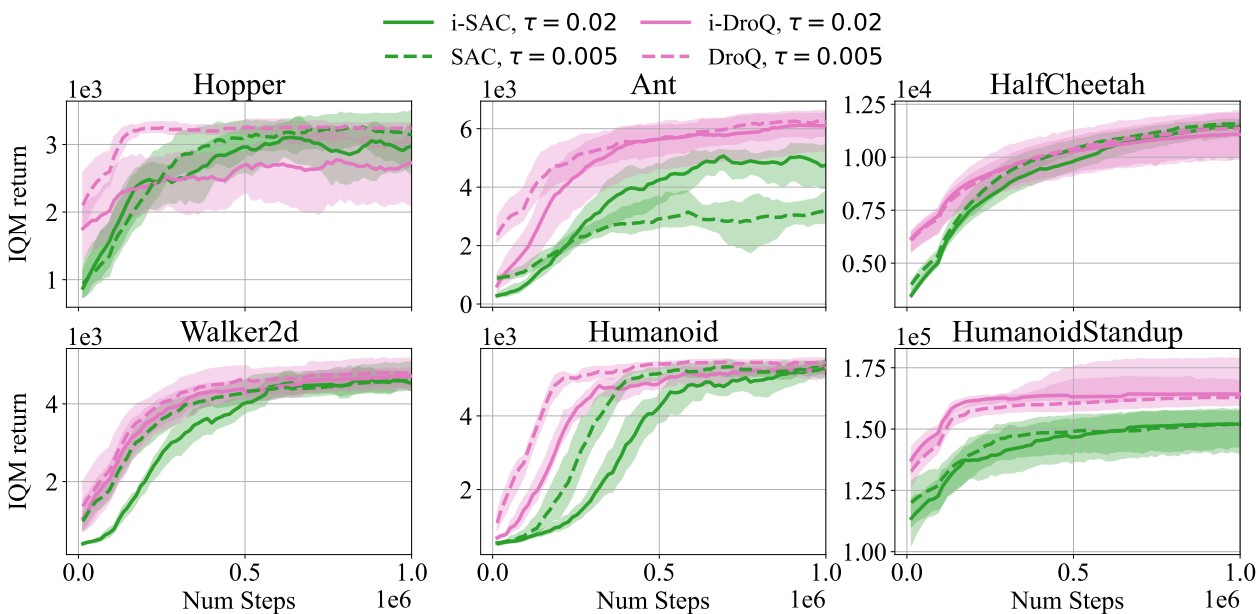

Figure 21: Training curves of i-SAC with $K = 4$, i-DroQ with $K = 4$ along with SAC and DroQ on 6 MuJoCo environments. In most problems, the iterated approach outperforms the sequential approach.

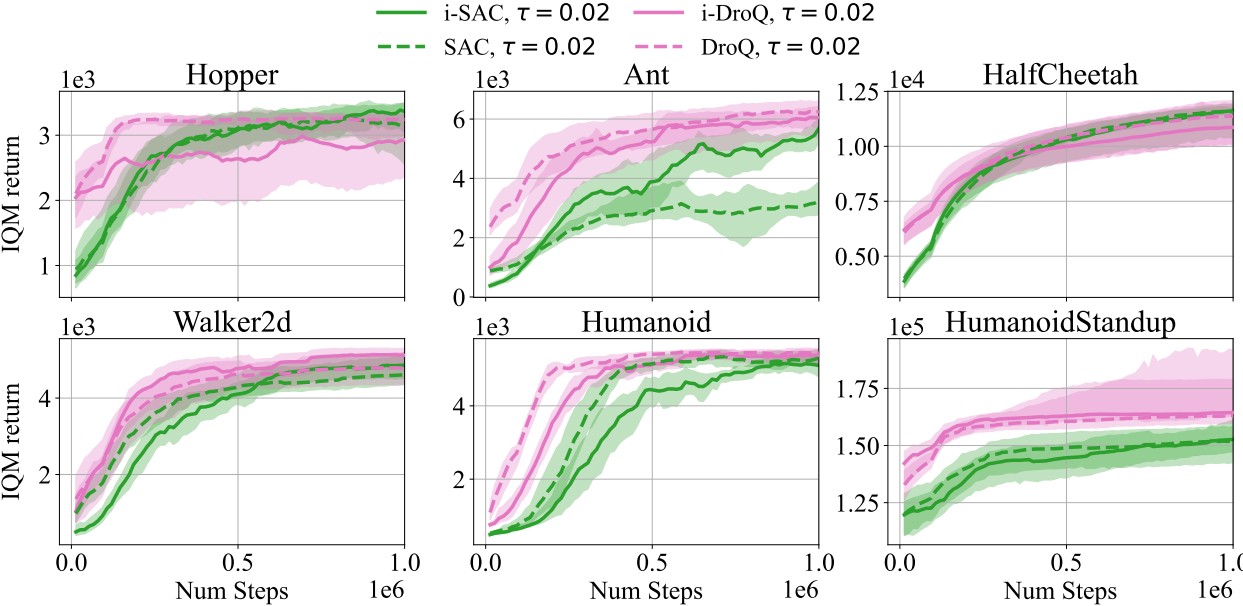

Figure 22: Training curves of i-SAC with $K = 4$, i-DroQ with $K = 4$ along with SAC and DroQ on 6 MuJoCo environments. Compared to Figure 21, all algorithms use the same soft target update frequency $\tau = 0.02$. In most problems, the iterated approach outperforms the sequential approach.

