# OpenReview forum: "Iterated $Q$-Network: Beyond One-Step Bellman Updates in Deep Reinforcement Learning"
_TMLR — Accepted by TMLR_

### Review · Reviewer_L6Jg · 2024-10-27

**Summary Of Contributions:**

The paper proposes a new way to successfully perform multiple iterations of Bellman updates at once. The authors proposed to replace the traditional 1-step Bellman updates of the form $(T \bar{Q} - Q)^2$ by a sequence of parallel updates where k Q-networks and targets are maintained. Each Q-network is updated based on the target of the previous Q-network and these updates can be performed in parallel and together. Authors demonstrate a theoretical condition when their method can help in reducing approximation errors, but in generally show through empirical results that their method can successfully leverage more gradient steps by increasing sample efficiency.

**Audience:**

Yes

**Claims And Evidence:**

Yes

**Requested Changes:**

1. Section 4 (the part before section 4.1) should be written with greater clarity and correctness. This would be crucial for recommending acceptance.

**Strengths And Weaknesses:**

Strengths:
1. The proposed method is simple and effective - a recipe that is important in scaling Deep RL. The method is generally applicable to any  Deep RL method, which uses critic updates, and adds an overhead of memory and time which scales linearly with the length of iterated updates.
2. The experiments are well designed to ablate where the benefit comes from when compared to basic DQN and IQN and variations on top of it.
3. The method gives a significant improvement in sample efficiency across both Atari and Mujoco domains.

Weaknesses:
1. Why are Steps 13 and 14 needed when there are Steps 11 and 12 in algorithm 1? Why is there a need to update the targets to current parameters rather than always updating to the online networks which uses the target parameters? If I understand correctly step 13 and 14, target Q_0 remains unupdated. Why   I felt the explanation of window size update with D and N was lacking and hard to understand. I think the second paragraph of Section 4 should be more clearly written to explain the core method of the paper clearly.
2. It seems in the algorithm, it is not made clear how $\hat{\theta}_0$ is initialized.
3. It is not clear to me how the theoretical result in Proposition 4.1 argues for decreasing the approximation error of full-gradient updates. Shouldn't the equation in line 2 of Page 5 should have $Q_{\theta_0}$ rather than the target notation? Can the assumptions of the proof  be explained more clearly?
4. I found the projection argument throughout the paper to be weak. Where is the projection argument used for? In all figures, it seems the projection is brining it back the targets back to the function space,but in Deep RL it is not clear we are limited by realizability in the function space. If the point is just to say that it takes a number of gradient updates for the function value to converge to the target value, then maybe that it might benefit to state that clearly.
5. One aspect of the experiment that remained unexplored was: How does the iterative Bellman update method compare to the methods which manage to successfully increase the UTD (Update to data) ratio? It seems they should be equivalent in terms of the empirical Bellman operator both methods are trying to perform. [1,2]


[1]: Chen, Xinyue, et al. "Randomized ensembled double q-learning: Learning fast without a model." arXiv preprint arXiv:2101.05982 (2021).
[2]: Hussing, Marcel, et al. "Dissecting Deep RL with High Update Ratios: Combatting Value Overestimation and Divergence." arXiv preprint arXiv:2403.05996 (2024).

---

> ### Author Response · Authors · 2024-12-18
>
> We thank the Reviewer for the insightful feedback and questions.
>
> ## Answer to the weaknesses
> > 1. a. Why are Steps 13 and 14 needed when there are Steps 11 and 12 in algorithm 1? Why is there a need to update the targets to current parameters rather than always updating to the online networks which uses the target parameters? [...]
>
> We argue that Steps $13$ and $14$ and Steps $11$ and $12$ are necessary as they have different purposes. On the one hand, Steps $13$ and $14$ are responsible for updating each target parameter to its respective online network to impose the structure of a chain. On the other hand, Steps $11$ and $12$ are responsible for updating each target parameter to the online parameter responsible for learning the *following* Bellman update. This makes the window shift forward. Figure $4$ (left) illustrates this difference, where each $\bar{Q}\_k$ is updated to $Q\_k$ every $D$ steps (Steps $13$ and $14$) and is updated to $Q\_{k+1}$ every $T$ steps (Steps $11$ and $12$). We point out that $D << T$ in practice, for example, $D=30$ and $T=6000$ in the Atari experiments.
>
> > 1. b. I think the second paragraph of Section 4 should be more clearly written to explain the core method of the paper clearly.
>
> We thank the Reviewer for their suggestion. In the updated submission, we modified the second paragraph of Section 4 to clarify the point raised by the Reviewer.
>
> > 2. It seems in the algorithm, it is not made clear how $\bar{\theta}_0$ is initialized.
>
> $\bar{\theta}_0$ and every $\theta_k$ are initialized randomly. This clarification has been added in the first paragraph of Appendix $C$ in the updated submission.
>
> > 3. a. It is not clear to me how the theoretical result in Proposition 4.1 argues for decreasing the approximation error of full-gradient updates. [...] Can the assumptions of the proof be explained more clearly?
>
> We first stress that the theoretical analysis is made to address the gap between theory and practice. This gap comes from the nature of the update as i-QN is a semi-gradient method (not a full-gradient method). This choice is made since minimizing one TD-error with semi-gradient updates is equivalent to minimizing the corresponding approximation error, when the dataset of samples is rich enough, as proven in Proposition $A.1$. This is why i-QN can make $K$ consecutive approximation errors decrease by learning $K$ consecutive Bellman updates. Unfortunately, in this setting, the sum of approximation errors is not guaranteed to decrease, even when each Bellman update is learned accurately such that each approximation error decreases. This comes from the fact that the target networks are frozen due to the semi-gradient updates. Therefore the constructed chain is broken: $Q\_{\bar{\theta}\_1} \neq Q\_{\theta_1}$, as illustrated in Figure $4$ (left). Nonetheless, Proposition $4.1$ presents a condition under which the sum of approximation errors is minimized. This condition states that the decrease of each approximation error should be greater than a specific quantity: the displacement of the targets, $|| \Gamma^* Q\_{\bar{\theta}\_{k-1}^{t+1}} - \Gamma^* Q\_{\bar{\theta}\_{k-1}^t}||, \forall k \in \{1, .., K\}$.
>
> We kindly ask the Reviewer to refer to the updated submission, where the paragraphs before Section $4.1$ are now rewritten in a clearer way.
>
> > 3. b. Shouldn't the equation in line 2 of Page 5 should have $Q_{\theta_0}$ rather than the target notation?
>
> $Q_{\theta_0}$ does not exist as the first $Q$-function is not learned. This is why $Q_{\bar{\theta}_0}$ is written with the target notation.
>
> > 4. I found the projection argument throughout the paper to be weak. Where is the projection argument used for? In all figures, it seems the projection is bringing it back the targets back to the function space, but in Deep RL it is not clear we are limited by realizability in the function space. If the point is just to say that it takes a number of gradient updates for the function value to converge to the target value, then maybe that it might benefit to state that clearly.
>
> We agree with the Reviewer that when very deep neural networks are used, the space of representable $Q$-functions $\mathcal{Q}_{\Theta}$ might cover the entire space of $Q$-functions $\mathcal{Q}$. However, very deep neural networks are not common in RL, as opposed to other fields like NLP or computer vision, especially for the actor-critic algorithms where most works use a $2$ hidden-layers MLP for the critics. This is why we include a projection operator in the figures.
>
> Nevertheless, we modified the footnote at the bottom of Page $3$ where we now specify that the presented abstraction in the figures still holds when the function approximators are powerful enough to cover the entire space of $Q$-functions.

---

> ### Author Response · Authors · 2024-12-18
>
> > 5. How does the iterative Bellman update method compare to the methods which manage to successfully increase the UTD (Update to data) ratio? It seems they should be equivalent in terms of the empirical Bellman operator both methods are trying to perform.
>
> We first stress that methods using high UTD are different than i-QN. Indeed, as opposed to i-QN, the additional gradient steps used by high UTD methods are not parallelizable. Moreover, the additional gradient steps used by high UTD methods are done on different samples which means that more samples are given to each $Q$-function as opposed to i-QN where the samples are shared across the additional gradient steps.
>
> We also recall that DroQ is a high UTD method as it uses a UTD of $20$. In Figure $9$, we show that combining i-QN with DroQ (i-DroQ) improves DroQ's performance.
>
> ## Requested Changes
> > Section 4 (the part before section 4.1) should be written with greater clarity and correctness.
>
> We thank the Reviewer for their honest feedback. We provide a new version of Section $4$ in the updated submission where modifications have been made w.r.t. the points raised by the Reviewer. For convenience, the modified text is written in orange.

---

> > ### Comment · Reviewer_L6Jg · 2025-01-04
> >
> > I thank the authors for their modifications and clarifications. I have gone through the paper again and have no questions for the authors at this point and my concerns are resolved.

---

### Review · Reviewer_7Lyx · 2024-11-12

**Summary Of Contributions:**

This paper introduces a novel architecture, called i-QN, for critics in
deep RL, wherein an ensemble of $K$ critics is trained so as to minimize
the error accumulated over $K$ applications of the Bellman operator. The
authors show that minimizing such a loss can lead to tighter upper
bounds on policy suboptimality in theory. Empirically, across a variety
of benchmarks, it is shown that i-QN consistently results in greater
performance or sample efficiency.

**Audience:**

Yes

**Broader Impact Concerns:**

No broader impact concerns

**Claims And Evidence:**

Yes

**Requested Changes:**

1.  In the abstract, "till now" is too informal, it should be "until
    now".
2.  The name "i-QN" collides with the already popular IQN
    algorithm/architecture (Implicit Quantile Network) – it may be best to consider a different
name.
1.  In the introduction, you describe the following as an efficiency
    issue in deep RL: "projection steps are made sequentially, thus forcing to learn
a Bellman update only at the end of the previous projection step". I
don't understand what this means—I think this could be spelled out and
motivated in a little more detail.
1.  I think it would be worth explicitly spelling out what you mean by
    the empirical estimate $\hat\Gamma$ of the Bellman operator. Readers that
haven't studied TD-learning in detail might not be comfortable with this
notion / might not realize that what you're describing here is the usual
approach to value-based deep RL. Likewise, there is more than one
empirical Bellman operator (e.g. the standard one vs that used in double
Q-learning). Similarly, I believe you should also cite some more
fundamental algorithms here, such as Q-Learning and DQN, which almost
all readers have likely implemented.
1.  The notation $Q^{\ast/\pi}$ is pretty odd. Since you often want to
simultaneously describe both Q-functions, it might be preferable to
write $Q^\bullet$ for $\bullet\in\{\ast,\pi\}$.
1.  There should be discussion comparing the proposed approach and
    methods such as Value Iteration Networks, which are closely related.
1.  At the beginning of section 4, it says "By learning one Bellman
    update at a time, classical $Q$-Network approaches minimize each term at a time". I
don't know that this is exactly correct. Q-learning approaches do not
minimize $\|\Gamma^\ast Q_{k-1} - Q_k\|$, they make indivual updates in
the direction minimizing those losses. It sounds like you're referring
more to something like FQI here. Maybe this is fine, since approaches
involving target critics are perhaps more akin to FQI than Q-learning,
but this should be explained.
1.  Why do you frequently write the sum of subsequent Bellman errors as
$\|\Gamma^\star Q\_{\overline{\theta}\_0} - Q\_{\theta\_1}\|^2 + \sum\_{k=2}^K\|\Gamma^\star Q\_{\overline{\theta}\_{k-1}} -
Q\_{\theta\_k}\|^2$ as opposed to a single term
$\sum\_{k=1}^K\|Q\_{\overline{\theta}\_{k-1}} - Q_{\theta\_k}\|^2$?
1.  In section 4, i-QN is discussed several times before it is actually
    defined. For example, at the bottom of page 4 it says "As i-QN is a semi-gradient
method…", but this is before i-QN has been described explicitly.
1.  The discussion just above section 4.1 is difficult to understand. It
    says "the optimization procedure is designed to make Equation 4 valid". Which
optimization procedure is this referring to? It would be helpful to
understand how this claim is true.
1.  There should be a discussion (and experiments?) comparing i-QN with
    something like Value Iteration Networks \[1; VIN\] and its descendants. Both attempt to
predict several iterations of Bellman updates. When should one prefer
i-QN to VIN and vice-versa? It appears that i-QN has the benefit of
enabling parallelized applications of the Bellman operator over its
heads, unlike VIN. If we could implement a "Transformer analogue of
VIN", would i-QN still retain any benefits?
1.  Likewise, there should be more discussion as well as experiments
    comparing i-QN with Bootstrapped DQN \[2\]. In the introduction, Bootstrapped DQN
is described as "\[improving\] the projection of $Q^{\ast/\pi}$
on $\mathcal{Q}_\Theta$". However, there is really more to it than that;
Bootstrapped DQN was designed to represent a posterior over Q-functions
for exploration with an ensemble. i-QN also introduces a form of
ensemble. Would Bootstrapped DQN perform worse than i-QN with the same
$K$? There are other subtle differences between Bootstrapped DQN and
i-QN. For example, in Bootstrapped DQN, you uniformly sample a
Q-function head at the beginning of each episode, and act greedily with
respect to it until the episode ends; in i-QN, you uniformly sample a
Q-function head at each *step* and act $\epsilon$-greedily. I would
think the Bootstrap DQN version could potentially be more useful for
(deep) exploration, especially if the $K$ heads in i-QN encode any form
of epistemic uncertainty. Do you think the improved performance of i-QN
can (at least in part) be due to improved exploration induced by the
disagreement between heads? This should be discussed in the manuscript.
1.  With regard to the IQM plots, it would be nice to see the suite of
statistics (e.g., optimality gap, etc) recommended by Agarwal et al. as
opposed to only IQM, which hides some useful information.
1.  Figure 6 middle uses colors that are very difficult to distinguish.
    It would also be helpful to include a reminder around this figure about what
$G, T$ are.
1.  In the "Gradient steps executed in parallel" part of section 6.1.1,
    you say that Figure 6 middle tests i-DQN with $K=4$ and $G=1$, but in the
Figure, it says i-DQN with $K=4$ and $G=4$. Which one is it? The latter
(setting described in the Figure) makes more sense to me in the context
of the experiment.

## References

1.  Tamar, Aviv, et al. "Value iteration networks." Advances in neural
information processing systems 29 (2016).
1.  Osband, Ian, et al. "Deep exploration via bootstrapped DQN."
    Advances in neural information processing systems 29 (2016).

**Strengths And Weaknesses:**

## Strengths

The i-QN architecture is a very interesting form of ensemble; rather
than training each ensemble member identically, the ensemble members act
as targets for a (ordered) *sequence* of Bellman targets. Notably, the
sequence of Bellman backups required for computing parameter updates can be
evaluated in parallel, so as to not dramatically increase training time.

The motivation for the i-QN architecture is nice, leveraging results
providing upper bounds on policy suboptimality. Empirically, the
performance of the various i-QN implementations appears to be
consistently good; I appreciated that the authors implemented i-QN
within a variety of base deep RL methods (e.g., including multiple
examples of value-based discrete-action methods and continuous control
actor-critic methods).

## Weaknesses

The main weakness of the paper, in my opinion, is a lack of comparison
with the most similar methods in the literature involving either an
ensemble of Q-functions or multiple applications of the Bellman
operator; most notably, Bootstrapped DQN and Value Iteration Networks
(see Requested Changes). More specifically, I believe the following
crucial experiments / discussions are missing:

1.  Comparison of i-QN to these existing architectures / methods;
2.  Study of which components are responsible for performance
    improvements (e.g. is it due to increased capacity? due to ensemble
    diversity? is it actually due to reduced error of accumulated
    Bellman errors?).

Moreover, as discussed in Requested Changes, some of the writing (particularly surrounding technical details) is imprecise and/or unclear.

---

> ### Author Response · Authors · 2024-12-18
>
> We thank the Reviewer for the thorough comments and questions.
>
> ## Answer to the weaknesses
> > 1. Comparison of i-QN to [Bootstrapped DQN and Value Iteration Networks].
>
> Thank you for this suggestion. We kindly ask the Reviewer to refer to Point $11$ for the comparison to Value Iteration Networks and to Point $12$ for the comparison to Bootstrapped DQN.
>
> > 2. Study of which components are responsible for performance improvements (e.g. is it due to increased capacity? due to ensemble diversity? is it actually due to reduced error of accumulated Bellman errors?).
>
> We argue that our work already contains ablations showcasing the benefit of each component of i-QN. We first recall that i-QN's main benefit is to minimize the sum of approximation errors. We empirically validate this claim in Figure $5$, where i-FQI yields a lower sum of approximation errors than FQI. These gains are correlated with better performances than FQI as Theorem $3.4$ in [1] suggests. We provide here an overview of the ablations analyzing i-QN's component:
>
> a. By learning multiple Bellman updates, i-QN distributes the samples across several $Q$-function in an efficient way. In Figure $6$ (middle), i-DQN with $K=4, G=4$ outperforms DQN with $G=1$ while each $Q$-network is trained with the same number of samples.
>
> b. By learning multiple Bellman udpates, i-QN paralelizes computations which yields higher performances earlier in the training. In Figure $6$ (right), after $25$M environment interactions, i-DQN with $K=4, T=2000$ reaches similar performances as DQN with $T=8000$ after $100$M environment interactions.
>
> c. The $Q$-networks trained by i-DQN are diverse as shown in Figure $18$. However, this diversity is not the core benefit of our method. Indeed, in Figure $8$ (left), we show that always sampling actions from the first or the last $Q$-network only brings a small decrease in performance.
>
> d. Even if i-QN uses more memory than classical QN approaches due to the need to store several $Q$-networks at once, i-QN's networks do not have a larger representation capacity. Indeed, each $Q$-network is composed of the same amount of parameters as the $Q$-network used in the sequential approach. Nonetheless, we demonstrate in Figure $17$ (right), that letting DQN use as many parameters as all Q-networks of i-DQN combined performs worse than i-DQN.
>
> e. The shared architecture of i-DQN allows for more weights updates of the shared layers than the layers of DQN. However, we show in Figure $17$ (left) that i-DQN with independent networks performs even better than i-DQN with shared layers.
>
> [1] Amir-massoud Farahmand. Regularization in reinforcement learning. 2011.

---

> ### Author Response · Authors · 2024-12-18
> **Requested Changes 1/3**
>
> > 1. In the abstract, "till now" is too informal, it should be "until now".
>
> We made the requested change in the updated submission.
>
> > 2. The name "i-QN" collides with the already popular IQN algorithm/architecture (Implicit Quantile Network) [...]
>
> We respectfully argue that i-QN does not collide with IQN as i-QN is a framework while IQN is an algorithm. Moreover, the hyphen helps distinguish one from the other.
>
> > 3. In the introduction, [...] "projection steps are made sequentially, thus forcing to learn a Bellman update only at the end of the previous projection step". I don't understand what this means [...].
>
> We thank the Reviewer for their feedback. We now explain this sentence further in the updated submission.
>
> > 4. a. I think it would be worth explicitly spelling out what you mean by the empirical estimate $\hat{\Gamma}$ of the Bellman operator. [...]
>
> We now explicitly write the expression the empirical estimate $\hat{\Gamma}$ in the last paragraph of Section $2$ in the updated submission.
>
> > 4. b. Similarly, I believe you should also cite some more fundamental algorithms here, such as Q-Learning and DQN, [...].
>
> We now cite Q-learning and DQN at this specific location in the updated submission.
>
> > 5. [...] it might be preferable to write $Q^{\bullet}$ for $\bullet \in \{*, \pi\}$.
>
> We thank the Reviewer for the suggestion. In the updated submission, we propose an alternative notation: $Q^{\star}$ which represents $Q^*$ in the AVI setting and $Q^{\pi}$ in the APE setting. Following this change in notation, we updated Figures $2, 3$, and $4$ (left).
>
> > 6. There should be discussion comparing the proposed approach and methods such as Value Iteration Networks, which are closely related.
>
> We kindly ask the Reviewer to refer to our answer of Point $11$ that concerns the same topic.
>
> > 7. At the beginning of section 4, it says "By learning one Bellman update at a time, classical $Q$-Network approaches minimize each term at a time". [...] this should be explained.
>
> As requested by Reviewer L6Jg, we reformulated Section $4$ by clarifying and explaining each statement. We have now added a justification for this specific statement.
>
> > 8. Why do you frequently write the sum of subsequent Bellman errors as $| \Gamma^* Q_{\bar{\theta}\_0} - Q\_{\theta_1}|^2 + \sum_{k=2}^K | \Gamma^*Q\_{\bar{\theta}\_{k-1}} - Q\_{\theta_k} |^2$
>
> > as opposed to a single term $\sum_{k=1}^K | \Gamma^*Q\_{\bar{\theta}\_{k-1}} - Q\_{\theta\_k} |^2$?
>
> We stress that the formulas are not the same in the submission as one contains target networks $\bar{\theta}_k$ and the other one contains online networks $\theta_k$. We break one sum in two terms as the first network, with index $0$, is not learned. This is why, we do not define its online version and only refer to the target parameters $\bar{\theta}_0$. In the third paragraph of Section $4$, we now clearly indicate that one sum is "without the target parameters".
>
> > 9. In section 4, i-QN is discussed several times before it is actually defined. [...]
>
> We believe that the updated version of Section $4$ now explains the proposed algorithm in a clear way.
>
> > 10. [...] It says "the optimization procedure is designed to make Equation 4 valid". Which optimization procedure is this referring to?
>
> The last paragraph before Section $4.1$ has been updated to further explain and justify this claim in more detail.

---

> > ### Author Response · Authors · 2024-12-18
> > **Requested Changes 2/3**
> >
> > > 11. a. There should be a discussion (and experiments?) comparing i-QN with something like Value Iteration Networks and its descendants. Both attempt to predict several iterations of Bellman updates. [...]
> >
> > We believe that VINs and i-QN's objectives differ. While i-QN's goal is to learn multiple Bellman updates concurrently, the goal of VINs is to train a policy end-to-end and to include planning steps in a finite MDP $\bar{\mathcal{M}}$ that is different from the original MDP $\mathcal{M}$ inside the policy architecture. This means that while i-QN's objective is to learn multiple Q-functions at once, VIN's objective is to learn the transition dynamics and the reward of the finite MDP $\bar{\mathcal{M}}$. This is why, the VIN framework entails some drawbacks that do not belong to the i-QN framework making the comparison impractical. Indeed, VINs do not scale well with the number of Bellman iterations (planning steps) needed for solving the task because the number of parameters used by VINs scales linearly with the number of planning steps as opposed to i-QN where the considered window of Bellman updates can be effectively shifted as explained in Section $4$. Additionally, as opposed to i-QN, VINs assume that, for each MDP $\mathcal{M}$, there exists a *finite* MDP $\bar{\mathcal{M}}$ such that the optimal plan for $\bar{\mathcal{M}}$ contains useful information about the optimal policy of the original MDP $\mathcal{M}$. This assumption is not guaranteed to hold for all MDPs and more importantly, some design choices need to be made to model $\bar{\mathcal{M}}$ in practice.
> >
> > For those reasons, we do not experimentally compare i-QN against VINs. Nonetheless, we added this discussion to Appendix $E$ of the updated submission.
> >
> > > 11. b. It appears that i-QN has the benefit of enabling parallelized applications of the Bellman operator over its heads, unlike VIN. If we could implement a "Transformer analogue of VIN", would i-QN still retain any benefits?
> >
> > We believe that a "Transformer analogue of VIN" goes against the core idea of VINs. Indeed, the core idea of VINs is to leverage the fact that CNNs can be used to encode one value iteration step implicitly. This is why, we argue that a "Transformer analogue of VIN" does not seem feasible at first glance. Moreover, we point out that each planning step of a VIN takes the value function obtained from the previous planning step as input. This is why the planning steps of VINs are not parallelizable as opposed to i-QN, where each $Q$-function takes a state and an action as input, therefore, all $Q$-functions can be evaluated in parallel.
> >
> > > 12. a. Likewise, there should be more discussion as well as experiments comparing i-QN with Bootstrapped DQN.
> >
> > > 12. b. [...] Would Bootstrapped DQN perform worse than i-QN with the same $K$?
> >
> > We thank the Reviewer for the suggestion. Bootstrapped DQN is a method that also uses an ensemble of $N$ $Q$-functions. However, Bootstrapped DQN and i-QN have orthogonal objectives: while Bootstrapped DQN focuses on deep exploration, i-QN aims at learning multiple Bellman updates concurrently. Nevertheless, in Figure $19$ of the updated submission, we propose to compare Bootstrapped DQN with $N = 6$ to i-DQN with $K = 5$ (i-DQN, head change for each sample) such that both algorithms use the same number of $Q$-functions. Indeed, i-DQN requires $K + 1$ networks. i-DQN outperforms Bootstrapped DQN on the $3$ considered Atari games.
> >
> > We remark that Bootstrapped DQN is unlearning on the game Asteroids as opposed to the two other games. We point out that in the original paper, the performances of Bootstrapped DQN are also low for this game. Additionally, the original performances correspond to a setting where sticky actions are not implemented, which might further decrease the performance of Bootstrapped DQN.

---

> > > ### Author Response · Authors · 2024-12-18
> > > **Requested Changes 3/3**
> > >
> > > > 12. c. [...] In Bootstrapped DQN, you uniformly sample a Q-function head at the beginning of each episode, and act greedily with respect to it until the episode ends; in i-QN, you uniformly sample a Q-function head at each step and act $\epsilon$-greedily. I would think the Bootstrap DQN version could potentially be more useful for (deep) exploration, especially if the $K$ heads in i-QN encode any form of epistemic uncertainty. Do you think the improved performance of i-QN can (at least in part) be due to improved exploration induced by the disagreement between heads? [...]
> > >
> > > In Figure $19$ of the updated submission, we now evaluate a version of i-DQN where the same $Q$-function is kept for interacting with the environment (i-DQN, constant head along episode). This version of i-DQN performs slightly worse than the version of i-DQN where a head is sampled at each environment interaction (i-DQN, head change for each sample). This suggests that keeping the same head along an episode to hope for deeper exploration, as suggested by [1], is less efficient than sampling a head at each environment interaction to avoid learning the other heads passively as identified by [2].
> > >
> > > We also included this analysis in the second last paragraph of Appendix $E$ in the updated submission.
> > >
> > > [1] Ian Osband, Charles Blundell, Alexander Pritzel, and Benjamin Van Roy. Deep exploration via bootstrapped dqn. NeurIPS, 2016.
> > >
> > > [2] Georg Ostrovski, Pablo Samuel Castro, and Will Dabney. The difficulty of passive learning in deep reinforcement learning. NeurIPS, 2021.
> > >
> > > > 13. With regard to the IQM plots, it would be nice to see the suite of statistics (e.g., optimality gap, etc) [...].
> > >
> > > We point out that the performance profile is presented in Figure $15$. In the updated submission, we complement our analysis by adding the optimality gap during training in Figure $15$ (right). Interestingly, i-DQN and i-IQN also improve on Atari games where human performance is not reached as their optimality gap is lower than their sequential counterpart.
> > >
> > > > 14. a. Figure 6 middle uses colors that are very difficult to distinguish.
> > >
> > > We point out that even if i-DQN $K=4, G=4$ is rendered in the same color as DQN $G=4$, the linestyle can be used to distinguish them. In the updated submission, we now changed the color of DQN, $G=1$ to brown instead of orange in Figure $6$.
> > >
> > > > 14. b. It would also be helpful to include a reminder around [Figure 6] about what $G, T$ are.
> > >
> > > We now recall the definition of $G$ and $T$ in the paragraph "Hyperparameter setting" of Section $6.1$.
> > >
> > > > 15. [...] you say that Figure 6 middle tests i-DQN with $K=4$ and $G=1$, but in the Figure, it says i-DQN with $K=4$ and $G=4$. Which one is it? [...]
> > >
> > > Thank you. We now corrected this mistake in the first paragraph of Section $6.1.1$.

---

> > > > ### Comment · Reviewer_7Lyx · 2025-01-13
> > > >
> > > > Thanks to the authors for their very detailed responses. I very much appreciated the discussion and experimental results added in the revision, my concerns are largely addressed.

---

### Review · Reviewer_T2xJ · 2024-12-12

**Summary Of Contributions:**

The authors introduced iterated Q-Network (i-QN) to learn multiple consecutive Bellman updates. The authors demonstrate the effectiveness of i-QN on the mujoco and atari benchmarks. Furthermore, they show that their algorithm is theoretically grounded.

**Audience:**

Yes

**Broader Impact Concerns:**

Not applicable since this work is theoretical in nature.

**Claims And Evidence:**

Yes

**Requested Changes:**

* Please include a discussion of computational complexity, FLOPS, etc.
* Ideally, you should keep the number of parameters constant. Specifically, is it better to scale the number of networks or having a bigger network?

Minor:
* Figure 6: The colors of DQN, G=1 and DQN, G=4 are difficult to distinguish.

**Strengths And Weaknesses:**

Strengths:
* The contribution is novel.
* The algorithm is simple and easy to implement.
* The algorithm is theoretically grounded.
* Ablation study takes into consideration the number of gradient steps, which network is used for action sampling, etc.
* Appropriate experimental settings i.e., both mujoco and atari.
* Use the baselines from the well established dopamine library.

Weaknesses:
* Ablation studies do not take into consideration computational complexity nor parameter counts.

---

> ### Author Response · Authors · 2024-12-18
>
> We thank the Reviewer for their insightful comments. As the weakness raised by the Review is also mentioned in the requested changes, our answers focus on the requested changes.
>
> > Requested change 1: Please include a discussion of computational complexity, FLOPS, etc.
>
> We thank the Reviewer for the suggestion. In the updated submission, we now report the additional FLOPs required by i-QN in Table $5$ in Appendix $D$. We observe that while the additional FLOPs required by i-QN for a gradient update scale linearly with $K$, sampling an action from an i-QN agent takes the same amount of FLOPs than sampling an action from a QN agent because each $Q$-function has the same architecture.
>
> > Requested change 2: Ideally, you should keep the number of parameters constant. Specifically, is it better to scale the number of networks or having a bigger network?
>
> We first point out that even if i-QN approaches require more parameters to be stored in the vRAM (see the second line of Table $5$ in Appendix $D$), each Q-function is composed of the same number of parameters. This means that the representation capacity of i-QN's $Q$-functions is the same as the $Q$-functions used by sequential approaches. Therefore, during inference, the number of parameters used to act on the environment is the same as for sequential approaches (see the fourth line of Table $5$ in Appendix $D$).
>
> Nonetheless, we kindly ask the reviewer to refer to Appendix $E$, Paragraph *i-DQN vs. DQN with an inflated number of parameters*, where we compare i-DQN to a version of DQN with an inflated number of parameters to match the number of parameters used by i-DQN. Figure $17$ (right) reports that the inflated version of DQN yields lower returns than i-DQN on the $2$ considered Atari games. This supports the idea that i-QN’s benefit lies in the way the networks are organized to minimize the sum of approximation errors.
>
> > Requested change 3: Figure 6: The colors of DQN, G=1 and DQN, G=4 are difficult to distinguish.
>
> We now changed the color of DQN, $G=1$ to brown instead of orange in Figure $6$.

---

### Author Response · Authors · 2025-02-05
**Camera Ready Version**

We thank the Action Editor and all Reviewers for their feedback and suggestions, which helped improve our work. We have de-anonymized our submission and made our code and models open source, as promised during the rebuttal.

---

### Decision · Action_Editor_u9u4 · 2025-01-21

**Recommendation:** Accept as is

**Comment:**

The reviewers are unanimous in agreeing this paper should be accepted as is, which I agree with. It is well-written, well-motivated, and provides adequate evidence for the claims.

**Audience:**

This paper would be of interest to most reinforcement learning researchers.

**Claims And Evidence:**

The authors demonstrate that their proposed algorithm, iterated Q-Network, is theoretically sound, and can be integrated into a variety of value-based algorithms with performance improvements. The authors additionally include ablation experiments which help provide a better understanding of their method.

The claims are well-supported by the evidence.